# Puerarin Modulates Hepatic Farnesoid X Receptor and Gut Microbiota in High-Fat Diet-Induced Obese Mice

**DOI:** 10.3390/ijms25105274

**Published:** 2024-05-12

**Authors:** Ching-Wei Yang, Hsuan-Miao Liu, Zi-Yu Chang, Geng-Hao Liu, Hen-Hong Chang, Po-Yu Huang, Tzung-Yan Lee

**Affiliations:** 1Graduate Institute of Clinical Medical Sciences, College of Medicine, Chang Gung University, Taoyuan 33302, Taiwan; ycw0426@gmail.com; 2Division of Internal and Pediatric Chinese Medicine, Center for Traditional Chinese Medicine, Chang Gung Memorial Hospital, Linkou 333423, Taiwan; 3Graduate Institute of Traditional Chinese Medicine, School of Chinese Medicine, College of Medicine, Chang Gung University, Taoyuan 33302, Taiwan; miaowhale@gmail.com; 4Department of Traditional Chinese Medicine, Chang Gung Memorial Hospital, Keelung 20401, Taiwan; changzhi887@gmail.com; 5School of Traditional Chinese Medicine, College of Medicine, Chang Gung University, Taoyuan 333323, Taiwan; ghliu.wis@gmail.com; 6Division of Acupuncture and Moxibustion, Center for Traditional Chinese Medicine, Chang Gung Memorial Hospital, Taoyuan 333423, Taiwan; 7Sleep Center, Chang Gung Memorial Hospital, Taoyuan 333008, Taiwan; 8Graduate Institute of Integrated Medicine, China Medical University, Taichung 40402, Taiwan; tcmchh55@gmail.com; 9Department of Chinese Medicine, Linsen Chinese Medicine and Kunming Branch, Taipei City Hospital, Taipei 10844, Taiwan

**Keywords:** puerarin, farnesoid X receptor, gut microbiota, obese, mitochondrial function, mitophagy

## Abstract

Obesity is associated with alterations in lipid metabolism and gut microbiota dysbiosis. This study investigated the effects of puerarin, a bioactive isoflavone, on lipid metabolism disorders and gut microbiota in high-fat diet (HFD)-induced obese mice. Supplementation with puerarin reduced plasma alanine aminotransferase, liver triglyceride, liver free fatty acid (FFA), and improved gut microbiota dysbiosis in obese mice. Puerarin’s beneficial metabolic effects were attenuated when farnesoid X receptor (FXR) was antagonized, suggesting FXR-mediated mechanisms. In hepatocytes, puerarin ameliorated high FFA-induced sterol regulatory element-binding protein (SREBP) 1 signaling, inflammation, and mitochondrial dysfunction in an FXR-dependent manner. In obese mice, puerarin reduced liver damage, regulated hepatic lipogenesis, decreased inflammation, improved mitochondrial function, and modulated mitophagy and ubiquitin-proteasome pathways, but was less effective in FXR knockout mice. Puerarin upregulated hepatic expression of FXR, bile salt export pump (BSEP), and downregulated cytochrome P450 7A1 (CYP7A1) and sodium taurocholate transporter (NTCP), indicating modulation of bile acid synthesis and transport. Puerarin also restored gut microbial diversity, the *Firmicutes*/*Bacteroidetes* ratio, and the abundance of *Clostridium celatum* and *Akkermansia muciniphila*. This study demonstrates that puerarin effectively ameliorates metabolic disturbances and gut microbiota dysbiosis in obese mice, predominantly through FXR-dependent pathways. These findings underscore puerarin’s potential as a therapeutic agent for managing obesity and enhancing gut health, highlighting its dual role in improving metabolic functions and modulating microbial communities.

## 1. Introduction

Obesity increases the risk of cardiovascular events and is associated with major risk factors for atherosclerosis, diabetes, metabolic syndrome, and other conditions often linked to nonalcoholic fatty liver disease (NAFLD) [1,2]. The excess accumulation of fat in liver cells can lead to hepatocellular injury through several mechanisms such as direct cellular toxicity caused by free fatty acids (FFAs), oxidative stress and lipid peroxidation, mitochondrial dysfunction, and cytokine-induced hepatotoxicity [3]. Hyperinsulinemia resulting from insulin resistance enhances the activity of sterol regulatory element binding protein 1c (SREBP1c), a transcription factor regulating genes involved in de novo lipogenesis [4,5]. This creates a vicious cycle of augmented hepatic de novo lipogenesis and increased delivery of free fatty acids to the liver from insulin-resistant adipose tissue [6,7]. Increased intake of simple carbohydrates enhances de novo lipogenesis, whereas saturated fats and cholesterol directly deliver lipotoxic intermediates to the liver [8]. Anti-oxidant deficiencies may also play a role by exacerbating oxidative stress [9].

The farnesoid X receptor (FXR) is a nuclear receptor highly expressed in the liver and intestine that plays a key role in maintaining bile acid homeostasis [10]. In obesity, FXR dysfunction contributes to the development of metabolic disorders including NAFLD and type 2 diabetes [11]. Obesity is associated with qualitative and quantitative changes in bile acid profiles due to altered biosynthesis, transport, and microbial biotransformation. Elevated levels of tauro-beta-muricholic acid (TβMCA)—a FXR antagonist—inhibit FXR activity and suppress expression of genes involved in bile acid synthesis and transport [12]. This leads to dysregulation of lipid and glucose metabolism. On the other hand, gut microbiota plays a pivotal role in bile acid metabolism and FXR signaling [13]. Obesity alters the composition of gut microbiota, decreasing bile-acid modifying bacteria like *Bacteroides*. This reduces levels of the FXR agonist tauro-alpha-muricholic acid (TαMCA) [10]. The altered bile acid composition exacerbates FXR inhibition, causing further metabolic dysfunction. In addition, obesity is associated with increased intestinal permeability (“leaky gut”), which allows bacteria and bile acids to enter the portal circulation. This hepatic exposure to high concentrations of bile acids and endotoxins contributes to FXR inhibition, inflammation, and liver injury. Overall, the complex interplay between bile acids, gut microbiota, and FXR signaling is significantly disturbed in obesity. Therapies that restore normal bile acid profiles and gut microbial communities may have potential for treating obesity-associated conditions like NAFLD by reactivating hepatic FXR. Further research is needed to better elucidate these complex mechanisms.

Puerarin, a major isoflavone glycoside derived from the root of the *Pueraria lobata* plant, has displayed a wide spectrum of pharmacological activities. This compound possesses remarkable potential in various aspects of health and well-being. One of the fundamental attributes of puerarin is its capacity to scavenge free radicals and inhibit lipid peroxidation, thus affording protection to cells against oxidative damage [14]. This anti-oxidative action can be attributed to the phenolic hydroxyl group structure present in puerarin [15]. Beyond its anti-oxidative properties, puerarin exhibits significant anti-inflammatory effects. It effectively suppresses the production of pro-inflammatory cytokines, including tumor necrosis factor alpha (TNF-α), interleukin (IL)-1β, and IL-6, while also inhibiting nuclear factor kappa B (NF-kB) activation [6,16]. In the cardiovascular arena, puerarin has demonstrated its prowess by enhancing myocardial contractility, improving coronary blood flow, and safeguarding cardiovascular tissue from ischemia/reperfusion injury. Additionally, it displays anti-hypertensive effects, further underscoring its relevance in cardiovascular health [17]. Finally, puerarin demonstrates intriguing anti-cancer properties, including anti-proliferative and pro-apoptotic activities against various cancer cell lines. Its potential to modulate multiple signaling pathways suggests it may be effective in suppressing tumor growth [18]. Puerarin emerges as a versatile and multi-target agent with documented anti-oxidant, anti-inflammatory, cardioprotective, neuroprotective, anti-diabetic, anti-osteoporotic, and anti-cancer effects, substantiated by cellular and animal models [19].

These mechanisms suggest its potential in preventing diseases associated with obesity-related diseases and the protective potential of puerarin is also noteworthy, as it can shield cells from damage induced by multiple aspects of biological dysfunctions. This multi-faceted approach suggests its potential as an adjuvant treatment for NAFLD. The potential of puerarin to advance the field of natural medicine and improve the management of various chronic diseases warrants continued investigation. Although suppression of pro-inflammatory and anti-oxidant signaling pathways may be involved in the pathogenesis of lipotoxicity, there is currently no evidence that puerarin acts as an anti-lipogenic agent against free fatty acid (FFA)-induced mitochondrial dysfunction in the liver.

The overall objectives of this study were to examine whether puerarin has a protective effect against high FFA-induced lipotoxicity in the liver and to explore a potential mechanistic link between the FXR and microbiota dysbiosis. We showed that HFD-induced increased lipogenesis signaling and mitochondrial dysfunction in obese mice. Following puerarin treatment, enhanced mitochondrial biogenesis and FXR regulation contributed to related microbiota composition. Attenuated inflammasome signaling and reduced loss of mitophagy may have also contributed to the protective action of puerarin.

## 2. Results

### 2.1. Puerarin Ameliorates HFFA-Induced Hepatic Lipid Accumulation, Mitochondrial Dysfunction, and FXR Signaing

We evaluated the impact of puerarin on alpha mouse liver 12 (AML12) cells following treatment with high free fatty acids (HFFA). Our findings indicate that puerarin effectively mitigates HFFA-induced lipid accumulation, as well as the expression of SREBP1, nuclear factor kappa-light-chain-enhancer of activated B cells (NF-κB), and reactive oxygen species (ROS) (Figure 1A,B). Concurrently, it improves mitochondrial morphology and reduces swelling (Figure 1B). High doses of puerarin increased the expression of FXR and bile salt export pump (BSEP) and decreased cytochrome P450 family 7 subfamily a member 1 (CYP7A1) expression in AML12 cells post-HFFA treatment (Figure 1C).

### 2.2. Puerarin Modulation of HFFA-Mediated Hepatotoxicity in siFXR AML12 Cells

Further experimentation involving siRNA-mediated FXR knockdown (siFXR) in AML12 cells revealed that siFXR markedly upregulated the expression of SREBP1, NF-κB, ROS, and CYP7A1 expression (Figure 2A,B), conversely, there was a significant reduction in mitochondria, FXR and BSEP expression (Figure 2A–C). After high-concentration free fatty acid treatment, there were no significant changes in FXR and BSEP expression (Figure 2C). Lipid accumulation, NF-κB expression, and mitochondrial damage significantly increased following HFFA treatment in the siFXR-treated group. High dose puerarin was significantly alter hepatic lipid accumulation, mitochondrial dysfunction, and FXR signaling expression in siFXR-treated AML12 cells. However, puerarin effectively alleviates HFFA-induced hepatic lipid accumulation, mitochondrial dysfunction, and FXR signaling, with its beneficial effects being more pronounced in control cells compared to siFXR-treated AML12 cells (Figure 1). Taken together, the beneficial effects of puerarin on hepatic lipid accumulation and mitochondrial dysfunction were dependent on FXR expression. This highlights the pivotal role of FXR in mediating the therapeutic benefits of puerarin on hepatic functions, underscoring the potential of targeting FXR signaling pathways in the treatment of hepatic disorders.

### 2.3. Puerarin Alleviates HFFA-Induced Inflammation and Mitochondrial Dysfunction in Hepatocytes

In order to evaluate the effect of puerarin on the concentration of pro-inflammatory cytokines in AML12 hepatocyte cultures induced by high concentrations of fatty acids, we analyzed the levels of interferon gamma (IFNγ), tumor necrosis factor alpha (TNFα), interleukin-1 beta (IL-1β), and monocyte chemoattractant protein 1 (MCP-1). The results showed that high concentrations of puerarin significantly reduced the levels of TNFα, IFNγ, MCP-1, and IL-1β induced by high concentrations of fatty acids. In AML12 cell lines treated with siFXR, the concentrations of TNFα, IFNγ, MCP-1, and IL-1β showed no significant difference compared to the control group, but the expression of pro-inflammatory factors significantly increased after treatment with HFFA. Moreover, the levels of pro-inflammatory factors in siFXR cells treated with HFFA were significantly higher than in the control group (Figure 3A–D). Compared to control cells, levels of ROS and MDA significantly increased after HFFA treatment. Furthermore, total anti-oxidant activity in HFFA-AML12 cells was significantly lower compared to the control group (Figure 3E,F). Puerarin significantly inhibited the production of ROS and malondialdehyde (MDA), providing direct evidence of its ability to mitigate HFFA-induced cytotoxicity. Compared to the control group, puerarin also demonstrated dose-dependent increases in anti-oxidant activity in HFFA-AML12 cells (Figure 3K).

To determine whether the reduction in cytotoxicity in HFFA-AML12 cells in the presence of puerarin was due to changes in mitochondrial and nuclear integrity, we stained puerarin-treated cells with JC-1 dye. Puerarin prevented mitochondrial and DNA damage caused by changes in mitochondrial membrane potential (ΔΨm) due to HFFA, one of the early events leading to functional changes and impaired mitochondrial integrity. The earliest processes leading to functional changes and mitochondrial integrity impairment are changes in mitochondrial membrane potential (ΔΨm), and puerarin can prevent these changes. We assessed the activity of mitochondrial oxidative phosphorylation (OXPHOS), which is crucial for various aspects of cellular life in aerobic conditions. Mitochondrial compositions produced by AML12 cells were used to measure activities related to complexes I, II, III, and IV. After HFFA treatment, our data showed that the specific activities of complexes I, II, III, and IV were significantly reduced (Figure 3I–L). Puerarin reversed the decline in OXPHOS enzyme activity. Additionally, we confirmed that compared to the control group, the expression of sirtuin 1 (*Sirt1*), peroxisome proliferator-activated receptor-γ coactivator 1-α (*Pgc1α*), uncoupling protein 1 (*Ucp1*), and mitochondrial transcription factor A (*Tfam*) mRNA were reduced in HFFA-AML12 cells (Figure 3M,P). Puerarin treatment enhanced the expression of markers of mitochondrial biogenesis and function in HFFA-AML12 cells. However, treatment with HFFA in siRNA cell lines led to a significant reduction in the expression of mitochondrial biogenesis and function markers, and this could not be restored to control levels after puerarin treatment. From these results, it is inferred that puerarin may improve pro-inflammatory factors and mitochondrial activity and function via the FXR pathway. To sum up, puerarin has shown its potential to alleviate HFFA-induced inflammation, reduce oxidative stress, maintain mitochondrial integrity, and improve mitochondrial function in hepatocytes possibly through modulation of the FXR pathway.

### 2.4. Puerarin Improves Liver Damage and Lipid Accumulation in HFD-Induced Obesity Mice

To investigate the efficacy of puerarin in mitigating liver damage and lipid imbalance induced by a HFD in obese mice, a comprehensive analysis was conducted. These findings suggest that puerarin effectively ameliorated obesity and corrected lipid metabolism. When compared to normal mice, FXR-deficient mice had a slightly lower body weight of approximately 3–5 g. HFD feeding significantly increased the body weight of FXR-deficient mice; however, administration of high-dose puerarin resulted in weight reduction, though not by a significant amount (Figure 4A). Body weight, food intake, ratio of liver weight to body weight, plasma alanine aminotransferase (ALT), hepatic histological scores, as well as plasma and hepatic triglyceride (TG) and free fatty acid (FFA) levels (Figure 4A–H) exhibited significant reductions in HFD-fed mice following puerarin treatment. In normal mice, hepatic cells displayed well-organized arrangements, whereas HFD-afflicted mice exhibited disordered hepatocyte arrangement, hypertrophy, fatty vacuoles, and infiltration of pro-inflammatory cytokines. Treatment with puerarin resulted in a more organized hepatocyte arrangement and alleviated liver damage (Figure 4E). In FXR-knockout (KO) mice, hepatocyte hypertrophy and fatty vacuoles were also observed. HFD feeding exacerbated hepatocyte hypertrophy, lipid vacuolization, and inflammatory cytokine infiltration, with no significant changes observed following puerarin treatment. We conducted a histological scoring of the HFD group, revealing significantly increased scores. Puerarin treatment significantly improved hepatic histological scores in normal mice with HFD but did not restore the scores in FXR KO mice to the levels observed in normal mice (Figure 4E,F and Table 1). In FXR KO mice, hepatic histology showed evident lipid vacuoles and lipid accumulation compared to normal mice. Following HFD feeding, lipid accumulation increased significantly and puerarin treatment did not significantly reduce lipid accumulation. To further elucidate the regulatory effect of puerarin on hepatic lipogenesis in HFD-fed mice, we analyzed key factors in the hepatic lipogenesis pathway. In comparison to normal mice, HFD-fed mice exhibited significant increases in the expression of SREBP-1c and cluster of differentiation 36 (CD36) proteins in the liver, both of which were notably reduced following puerarin treatment in normal mice (Figure 4I–K). However, in FXR KO mice, puerarin treatment did not lead to a significant reduction in the expression of these proteins (Figure 4I). Furthermore, mRNA expression of the target genes *Srebp1*, fatty acid synthase (*Fas*), stearoyl-CoA desaturase-1 (*Scd1*), and acetyl-CoA-1 carboxylase (*Acc1*) was also decreased by puerarin treatment in HFD-fed mice (Figure 4L–O). However, in FXR KO mice, puerarin treatment did not lead to a significant reduction in the expression of these proteins and mRNAs (Figure 4L–O). Therefore, our results demonstrate that puerarin effectively reduces hepatic lipid accumulation and regulates the hepatic lipogenesis pathway. However, its effectiveness in improving hepatic histology is more pronounced in normal mice compared to FXR knockout mice, suggesting a potential role of FXR in mediating some of puerarin’s effects on liver damage and lipid metabolism in the context of HFD-induced obesity.

### 2.5. Puerarin Suppresses Inflammatory Response in Liver of HFD-Induced Obesity Mice

The activation of the NLR family pyrin domain containing 3 (NLRP3) inflammasome and the chemokine MCP-1 play crucial roles in the inflammation and immune response regulation in obese mice. To confirm whether puerarin mitigates obese inflammation by inhibiting NLRP3 inflammasome activation, we analyzed the expression of NLRP3 inflammasome-related factors and MCP-1. The protein expression of NLRP3 and MCP-1 significantly increased in the livers of normal mice fed a HFD, and the increase was significantly reduced following puerarin treatment (Figure 5A,B). The mRNA expression of *Tnfα*, *Ifnγ*, *Il-1β*, *Mcp-1*, inflammasome-related factors *Nlrp3*, *Pannexin*, apoptosis-associated speck-like protein containing a CARD (*ASC*), and pro-caspase-1 (*Pro-casp 1*) significantly increased in normal mice following a HFD and significantly decreased after puerarin treatment (Figure 5C,D). In FXR KO mice, pro-inflammatory factors and inflammasome-related factors were significantly higher than in normal mice. The protein and mRNA expression of pro-inflammatory and inflammasome-related factors significantly increased after HFD treatment and were not significantly improved by puerarin treatment. These results confirm that puerarin reduces the inflammatory response in the liver HFD-induced obese, and FXR plays a crucial role in the pro-inflammatory and inflammasome mechanisms.

### 2.6. Regulation of Mitochondrial Biogenesis by Puerarin in HFD-Induced Obesity Mice

To determine whether puerarin is beneficial for mitochondrial dysfunction induced by a high-fat diet in obesity, we analyzed mitochondrial biogenesis and mitophagy, both of which are essential for the pathogenic mechanism improvement in obesity. We confirmed that the protein and mRNA expression of SIRT1, mitochondrial complexes, PGC1α, and UCP1 was significantly reduced in the livers of mice fed a HFD (Figure 6A,B), and puerarin significantly increased the expression of mitochondrial biogenesis makers. In FXR KO mice, there was no significant difference in mitochondrial biogenesis factors and mitochondrial complex activity compared to normal mice. Subsequent HFD treatment led to a significant decrease in mitochondrial biogenesis factors and mitochondrial complex activity, with a significant increase observed after puerarin treatment. However, the improvement in mitochondrial biogenesis factors and mitochondrial complex activity by puerarin in FXR KO mice fed a HFD was limited and could not fully restore the levels of normal mice (Figure 6). Additionally, puerarin significantly increased anti-oxidant activity and reduced MDA expression, promoting mitochondrial function in fatty liver induced by a HFD (Figure 6D,E). This study’s findings indicate that puerarin has a positive impact on mitochondrial biogenesis in obese mice. However, under certain conditions, its effectiveness may be limited. These results also confirm that puerarin can improve the mitochondrial biogenesis pathway through the activation of FXR.

### 2.7. Regulation of Mitophagy and the Ubiquitin-Proteasome System by Puerarin in Obese Mice

To further explore the regulatory effects of puerarin on mitophagy and the ubiquitin-proteasome system in the liver of obese mice, we conducted analyses using both normal and FXR KO mice subjected to a HFD. The results indicated that the HFD led to a significant decrease in mitochondrial autophagy factors such as Parkin (Figure 7A), as well as in the mRNA expression of *Pink*, *Ndp52*, prohibitin 2 (*Phb2*), *Ambra1* interacting protein 3 (*Binp3*), *Nix*, *Fundc1*, and *Bcl2* (Figure 7B,C). Treatment with puerarin significantly improved the expression of these mitophagy factors. Puerarin also significantly increased the protein expression of ubiquitin and LC3B in fatty liver (Figure 7A). In FXR KO mice, the expression of mitochondrial autophagy and ubiquitin proteins showed no significant difference compared to normal mice. However, the HFD resulted in a significant reduction in the expression of mitophagy factors and ubiquitin proteins, and puerarin treatment was unable to significantly restore the expression of mitophagy and ubiquitin proteins to the levels observed in normal mice (Figure 7A–C). These results suggest that the regulation of mitochondrial autophagy factors and ubiquitin expression by puerarin in fatty liver requires modulation through FXR. Therefore, puerarin demonstrates regulatory effects on mitochondrial autophagy and the ubiquitin-proteasome system in the livers of obese mice, with its positive impact on mitophagy factors and ubiquitin expression contingent on FXR modulation underscoring its potential role in ameliorating hepatic dysfunction in HFD-induced obesity.

### 2.8. Impact of Puerarin on Bile Acid Transport Proteins in Obese Mice

To investigate the impact of puerarin on bile acid homeostasis in HFD-induced obese mice and the role of FXR signaling, this study assessed the expression of liver proteins involved in bile acid transport as well as the key enzyme in bile acid synthesis, CYP7A1, and examined factors related to bile acid metabolism. To investigate the influence of FXR deficiency on the FXR signaling pathway, downstream target factors were analyzed. In comparison to normal mice, FXR and BSEP expression significantly decreased in FXR-deficient mice, with FXR protein expression nearly absent (Figure 8). Conversely, CYP7A1 and sodium taurocholate co-transporting polypeptide (NTCP) expression increased significantly in FXR KO mice. Following HFD feeding, FXR and BSEP expression in the livers of FXR KO mice remained relatively unchanged, while CYP7A1 and NTCP expression increased significantly. Upon administration of puerarin, there were no significant changes in FXR, BSEP, or CYP7A1 expression in the livers of FXR-deficient mice, but NTCP expression showed a significant decrease, although it remained higher compared to normal mice. Taken together, FXR signaling appears to be critical in mediating these effects, emphasizing the potential of puerarin as a therapeutic agent in ameliorating hepatic dysfunction associated with obesity-induced alterations in bile acid metabolism.

### 2.9. Puerarin Modulates Gut Microbiota Composition in HFD-Induced Obesity Mice

To understand the role of gut microbiota in the development of obesity induced by a HFD and how puerarin might influence this relationship, we analyzed the gut microbiota composition. Principal component analyses (PCA) revealed that the gut microbiota in the HFD treated group deviated from the untreated group, and puerarin partially restored the level of gut microbiome in HFD-treated mice (Figure 9A). In addition, the gut microbiota profile of FXR KO group differed from the normal group. Alpha diversity indices, including observed Chao1 and Shannon’s diversity indices, were lower in the HFD-treated normal mice than in the untreated mice; however, the diversity was restored to a normal level by puerarin treatment (Figure 9B,C). The alpha diversity was reduced in FXR KO mice compared with normal mice, and it further decreased with HFD treatment. Notably, puerarin treatment did not alter the alpha diversity in these mice (Figure 9B,C). 

Puerarin markedly ameliorated the HFD-induced dysbiosis of the gut microbiota, evidenced by the restoration of the *Firmicutes*/*Bacteroidetes* ratio, as well as the relative abundances of the *Firmicutes* and *Proteobacteria* by phyla (Figure 9D–G). In FXR KO mice, there was an increase in the abundance of *Proteobacteria* along with a decrease in Firmicutes. A HFD significantly increased the abundance of *Proteobacteria* while significantly reducing *Firmicutes* in the intestines of FXR KO mice (Figure 9D–K). In FXR KO mice, puerarin did not induce significant changes in microbiota composition. Puerarin significantly restored the dysbiosis of gut microbiota caused by a HFD, including the abundance of *Clostridiaceae*, *Helicobacteraceae*, *Erysipelotrichaceae*, and *Porphyromonadaceae* by family in normal mice (Figure 9H–L). In FXR KO mice, there was an increase in the abundance of *Helicobacteraceae* and *Paraprevotellacea*, along with a decrease in *Clostridiaceae*, *Erysipelotrichaceae*, and *Porphyromonadaceae*. A HFD significantly increased the abundance of *Helicobacteraceae*, *Paraprevotellaceae*, but did not significantly change the abundance of *Clostridiaceae*, *Erysipelotrichaceae*, and *Porphyromonadaceae* in FXR KO mice (Figure 9H–L).

Upon further analysis of the microbial composition of species (Figure 9H), we observed that in normal mice subjected to a HFD, there was a notable increase in the levels of *Clostridium celatum* and *Helicobacter hepaticus*, accompanied by a decrease in *Akkermansia muciniphila* and *Turicibacter sanguinis*. Subsequent administration of puerarin post-HFD significantly reversed these changes, elevating the abundance of *Akkermansia muciniphila* and *Turicibacter sanguinis* to near-normal levels, while concurrently reducing the prevalence of *Clostridium celatum* and *Helicobacter hepaticus* (Figure 9N–Q). In FXR KO mice, the abundance of *Clostridium celatum*, *Akkermansia muciniphila*, and *Turicibacter sanguinis* significantly decreased, and there was no change in the abundance of *Helicobacter hepaticus* compared with normal (Figure 9N–Q). Notably, under HFD conditions in FXR KO mice, the abundance of *Helicobacter hepaticus* increased and that of *Akkermansia muciniphila* decreased, while *Clostridium celatum* and *Turicibacter sanguinis* showed no further changes. Puerarin treatment significantly reduced the abundance of *Clostridium celatum* and partially restore *Akkermansia muciniphila*, although not to the levels of normal mice (Figure 9N–Q). The abundances of *Clostridium celatum* and *Turicibacter sanguinis* were not further changed under HFD conditions with puerarin in FXR KO mice.

These findings suggest that FXR deficiency is associated with a pro-inflammatory gut microbiota profile, emphasizing the role of gut microbiota in fatty liver disease pathogenesis and the potential therapeutic impact of puerarin in modulating these effects. In summary, puerarin ameliorates HFD-induced alterations in gut microbiota, particularly in normal mice, while FXR deficiency leads to a distinct microbiota profile, underscoring the relevance of gut microbiota in fatty liver disease development and the potential of puerarin as a therapeutic intervention.

## 3. Discussion

The present study examined the effects of puerarin, an isoflavone with bioactive properties and gut microbiota composition, in murine models of obesity induced by HFD. Obesity is associated with alterations in FXR signaling and gut microbiota dysbiosis. Bariatric surgery has been shown to normalize FXR signaling and the gut microbiome in obese patients [21]. Animal studies have also demonstrated links between obesity, bile acids, and gut microbiota. For example, obese mice had more *Firmicutes* and less *Bacteroidetes* compared to lean mice. Treating the obese mice with bile acids led to weight loss [22]. In humans, obese individuals had lower secondary bile acid levels and different gut microbiome compositions compared to lean controls. Their study also showed that weight loss increased bacterial gene functions related to bile acid metabolism [23]. Taken together, these findings imply that obesity is associated with alterations in bile acid metabolism and gut microbiota dysbiosis. Strategies targeting the gut microbiome and bile acid signaling, such as bariatric surgery or bile acid treatment, may help normalize the biological changes associated with obesity and consequently promote weight loss. FXR signaling plays an important role in regulating bile acid metabolism, inflammation, and tumorigenesis in the liver. Specifically, FXR activation has been shown to inhibit lipogenesis and gluconeogenesis gene expression [24]. This is thought to occur through the suppression of the SREBP-1c pathway, as FXR signaling can exert anti-lipogenic effects in hepatocytes independent of SREBP signaling [25]. By regulating bile acid levels and inhibiting inflammatory pathways, FXR signaling may also suppress inflammation and tumorigenesis in the liver [25]. In summary, through its effects on lipid metabolism, inflammation, FXR signaling, and gut microbiota composition, puerarin serves as a key regulator of hepatic health and disease. We first illustrated that puerarin mitigated HFFA-induced hepatocyte damage by reducing inflammation, oxidative stress, and improving mitochondrial function. The beneficial effects were mediated, in part, through modulation of FXR signaling and gut microbiota composition. The proposed mechanisms may involve inhibiting the SREBP1 and NF-kB pathways, as well as regulating the mitochondrial biogenesis and dynamics. Previous studies have demonstrated the hepatoprotective effects of puerarin in other models of liver injury. For example, Zhang et al. showed that puerarin reduced liver injury in alcoholic fatty liver disease by attenuating oxidative stress and inflammation [26]. Wang et al. found that puerarin protected against hepatic fibrosis in rats by inhibiting TGF-β1 signaling and reducing inflammation and collagen deposition [27]. Additionally, Liu et al. demonstrated that puerarin inhibited NLRP3 inflammasome activation in mice with non-alcoholic steatohepatitis, consequently reducing inflammation and hepatocyte apoptosis [28]. Our current study provides novel evidence that puerarin protects hepatocytes specifically from HFFA-induced dysfunction through modulating FXR signaling and bile acid homeostasis. This adds to existing literature supporting the hepatoprotective effects of puerarin through reduction of inflammation and oxidative stress.

In addition, puerarin’s beneficial metabolic effects were attenuated when FXR was antagonized, suggesting FXR-mediated mechanisms. Supplementation with puerarin reduced plasma ALT, liver TG, and liver FFA in obese mice. These findings suggest that puerarin effectively ameliorated obesity and corrected lipid metabolism. FXR activation has been found to be protective against liver inflammation and injury in animal models [29]. FXR knockout mice had increased inflammation and inflammasome activation in the liver compared to wildtype mice. This suggests FXR normally acts to suppress inflammation [30]. Hepatic lipogenesis (fat synthesis in the liver) may contribute to liver inflammation by increasing lipid accumulation, oxidative stress, and activation of inflammatory pathways [31]. The inflammasome is a multiprotein complex that activates inflammatory responses, and there is evidence that it can promote liver inflammation and fibrogenesis when dysregulated [32]. In our current results, we confirm that puerarin reduces the hepatic inflammatory response in HFD-induced obese mice, which summarizes valid scientific findings on the anti-inflammatory effects of FXR in the liver and the pro-inflammatory effects of the inflammasome. Loss of FXR’s suppressive effects likely contributes to worse inflammation in FXR knockout mouse models.

Excessive hepatic lipogenesis can lead to fat accumulation (hepatic steatosis or fatty liver) and inflammation in the liver. The underlying mechanisms linking excessive lipogenesis to liver inflammation result in lipid accumulation in hepatocytes, which can directly induce inflammatory responses involving the release of pro-inflammatory cytokines and the activation of inflammatory pathways [33,34]. Fatty acid metabolism in the liver generates ROS. Increased ROS production leads to lipid peroxidation, mitochondrial damage, and oxidative stress [35]. Impaired mitochondrial function exacerbates ROS production and oxidative stress, further contributing to inflammation [36]. Accumulated lipids may directly activate inflammasomes, which are multi-protein complexes that promote inflammation [37]. Overall, dysregulated lipid metabolism triggers multiple pro-inflammatory mechanisms in the liver, including direct induction of inflammatory pathways, mitochondrial dysfunction, oxidative stress, and inflammasome activation. Puerarin maintaining proper lipid metabolism and mitochondrial health are important factors in preventing excessive inflammation associated with fat accumulation in the liver. Controlling hepatic lipogenesis can help restore balance and attenuate damaging inflammatory responses. In obese mice, puerarin reduced liver damage, regulated hepatic lipogenesis, decreased inflammation, improved mitochondrial function, and modulated mitophagy and ubiquitin-proteasome pathways. Subsequent HFD treatment led to a significant decrease in mitochondrial biogenesis factors and mitochondrial complex activity, with a significant increase observed after puerarin treatment. Puerarin significantly increased the expression of mitochondrial biogenesis-related markers and the improvement in mitochondrial biogenesis factors and mitochondrial complex activity by puerarin in mice fed a HFD could restore to the levels of normal mice. Our findings indicate that puerarin has a positive impact on mitochondrial biogenesis in obese mice. These results also confirm that puerarin can improve the mitochondrial biogenesis pathway through the activation of FXR. However, under FXR knockout conditions, its effectiveness may be limited.

FXR signaling plays an important role in HFD-induced disruption of bile acid homeostasis in obese mice. A study by Zhang et al. showed that activating FXR ameliorated HFD-induced obesity and insulin resistance in mice. They found that HFD suppressed intestinal and hepatic FXR signaling, leading to impaired bile acid homeostasis [24]. High-fat feeding in mice decreased the expression of FXR and its target genes involved in bile acid synthesis and transport [38]. FXR activation normalized bile acid levels and improved gut barrier function in a HFD mice [39]. FXR KO mice have disrupted bile acid regulation, which is dysregulated further by HFD. The composition of the bile acid pool is an important indicator of hepatic health. Loss of FXR disrupts bile acid regulation, which worsens with high-fat diet-induced obesity. Our results demonstrate that puerarin, a natural compound, modulated bile acid homeostasis in the liver by regulating the expression of key proteins involved in bile acid synthesis and transport. Specifically, puerarin upregulated the bile acid sensor FXR and the bile acid exporter BSEP. It downregulated CYP7A1 (the rate-limiting enzyme in bile acid synthesis) and NTCP (which transports bile acids into the liver). This indicates that puerarin reduces bile acid synthesis and increases bile acid excretion from the liver. On the other hand, the effects of puerarin on bile acid regulation require functional FXR signaling. FXR knockout mice showed disrupted bile acid homeostasis with decreased FXR/BSEP and increased CYP7A1/NTCP compared to normal mice. Feeding FXR KO mice a HFD, which causes obesity and metabolic dysfunction, further dysregulated bile acids. Puerarin only partially restored bile acid levels in FXR KO mice on a high-fat diet. Overall, these findings demonstrate that puerarin improves obesity-associated hepatic dysfunction by modulating bile acid metabolism signals in an FXR-dependent manner. FXR signaling is critical for puerarin’s effects on bile acid synthesis and transport.

Previously findings highlight the potential of targeting FXR pathways as a therapeutic strategy to mitigate gut inflammation and related pathologies between FXR, bile acids, and the gut microbiota particularly in understanding and treating gastrointestinal diseases [40]. A 2023 study by Wei et al. illustrated how intestinal FXR controls dysbiosis and liver injury in cholestatic mice [41]. They found FXR activation reshaped the gut microbiome and showed protective effects on the liver. This work implies that exploring FXR agonists could modulate the microbiome to improve cholestatic liver disease. Jiang et al. are currently studying how FXR signaling regulates anti-microbial peptide expression in the intestine and consequently alter microbiome composition. They are studying how this FXR-microbiota axis affects the progression of fatty liver disease [42]. The interactions between microbiome metabolites, bile acids, and FXR signaling in the progression of non-alcoholic fatty liver disease. This large collaborative effort hopes to uncover microbiome-based biomarkers of liver disease guided by FXR [43]. FXR agonists could shift the gut microbiome to increase production of beneficial primary bile acids in advanced liver disease [44]. This could potentially protect the liver by restoring bile acid homeostasis. In our results, the gut microbiota is an important player in mediating the effects of puerarin and progression of fatty liver disease. FXR plays a key role in maintaining normal gut microbiota composition. Loss of FXR leads to dysbiosis that worsens with HFD and cannot be corrected by puerarin without functional FXR signaling.

On the other hand, a probiotic which contains bacteria from the *Firmicutes* and *Actinobacteria phyla*, increased hepatic FXR signaling and altered bile acid profiles in a mouse model of cholestasis [42]. Liu et al. found that increased *Erysipelotrichaceae* in mice fed a HFD was associated with decreased FXR expression and activity [45,46]. Inagaki et al. showed *Proteobacteria*-derived lipopolysaccharide suppressed FXR activity and expression in mouse liver [47]. *Porphyromonadaceae* abundance negatively correlated with FXR signaling in patients with NAFLD [21]. Jia et al. reported that gut microbiota from the phyla *Firmicutes* and *Proteobacteria* metabolize primary bile acids into secondary bile acids like deoxycholic acid (DCA) and regulate bile acid homeostasis [48]. Liu et al. found increased *Erysipelotrichaceae* in mice fed a HFD, which was associated with decreased hepatic FXR signaling and altered bile acid composition [45]. *Clostridiaceae* abundance was positively correlated with secondary bile acids and liver inflammation in patients with NAFLD [49]. Puerarin partially restored the dysbiotic and the alpha diversity gut microbiota caused by HFD in normal mice. Specifically, puerarin normalized the abundance of certain microbe groups altered by HFD including *Firmicutes, Proteobacteria, Erysipelotrichaceae, Clostridiaceae, Porphyromonadaceae,* and *Helicobacteraceae*.

In this study, we found that puerarin significantly increased the abundance of Akkermansia muciniphila, aligning with the findings of Wang et al. Concurrently, Wang et al. demonstrated that puerarin alleviated intestinal barrier dysfunction and inflammation in mice with HFD-induced obesity, while also addressing related conditions such as steatosis and insulin resistance [50]. Known for its positive effects on diabetes and immune homeostasis, Akkermansia muciniphila plays a crucial role. *Clostridium celatum* is a member of the *Clostridia* class, a diverse assembly encompassing both commensal and pathogenic strains. Beneficial *clostridia* contribute to gut homeostasis and the fermentation of dietary fiber; however, certain species may behave as opportunists, fostering inflammatory states and dysbiosis when proliferating under conditions such as a high-fat diet (HFD) [51]. The observed increase in the abundance of *Clostridium celatum* and other members of the *Clostridia* class in response to a high-fat diet (HFD) could potentially activate inflammatory pathways and compromise gut integrity. This disturbance in gut barrier function and immune system engagement may exacerbate systemic inflammation, predisposing individuals to metabolic disorders such as obesity [51]. TCA significantly decreased the relative abundance of three culturable species of commensal bacteria, *Turicibacter sanguinis*, *Lactobacillus johnsonii*, and *Clostridium celatum*, in both cecal contents and mucosal scrapings from the colon [52]. Thus, shifts in *Clostridium* populations with the capacity to alter bile acid profiles could attenuate these FXR-driven protective effects, underscoring the multifaceted implications of gut microbiota composition in metabolic health.

Our findings align with previous studies demonstrating that *Helicobacter hepaticus* (*H. hepaticus*) induces chronic hepatitis and liver tumors in mice and is also associated with a significant increase in the prevalence of nonalcoholic fatty liver disease (NAFLD) [53,54]. In particular, mice on a high-fat diet showed increased *H. hepaticus* levels, which were reduced following treatment with PUR. Interestingly, this reduction was less pronounced in mice, suggesting that FXR plays a role in modulating the response to *H. hepaticus* infections under high-fat dietary conditions [55]. Our study provides valuable insights into the association among puerarin treatment, altered gut microbiota composition and diversity, increased FXR expression, and reduced steatosis. However, it is crucial to acknowledge some limitations. While we observed these associations, our study does not conclusively establish a causal relationship between the puerarin-induced changes in gut microbiota diversity, FXR expression, and steatosis inhibition. The precise mechanisms underlying these observations warrant further investigation. We hypothesize that the observed effects may be mediated through a similar mechanism, but further mechanistic studies are necessary to delineate the precise pathways involved. Additionally, our study focused specifically on the potential beneficial effects of puerarin in combating obesity by promoting favorable changes in the gut microbiota. However, the broader implications and applications of these findings, such as the role of puerarin in other metabolic disorders or the potential clinical applications, require further exploration and research.

## 4. Materials and Methods

### 4.1. Cell Culture

AML12 cells were seeded at 5 × 10^6^/dish in 10-cm culture dishes and used at 70–80% confluence. The cells were maintained at 37˚C with humidified air in a 5% CO_2_/95% air atmosphere. AML12 cells were incubated with HFFA, which were prepared according to a slightly modified method [56,57]. Puerarin was purchased from Sigma (St. Louis, MO, USA). Confluent cells were incubated in HFFA medium and with or without puerarin with the appropriate experimental conditions for the indicated time points.

### 4.2. siRNA Transfection

FXR-siRNA and negative control siRNA (5′-uguucuguuagcauaccuuTT-3′, Nippongene) (universal negative control siRNA, Nippongene, Tokyo, Japan) were transfected using Lipofectamine 2000 reagent (Invitrogen, Waltham, MA, USA) with a ratio of siRNA to Lipo2000 at 100 pmol:5 μL (for each well of a 6-well plate). Two concentration gradients of GCK-siRNA [100 nM (200 pmol siRNA) and 200 nM (400 pmol siRNA)] were used and incubated for 24 h. Lipofectamine RNAi-MAX (Thermo Fisher Scientific, Waltham, MA, USA) was used according to the manufacturer’s instructions.

### 4.3. Mitochondrial Analysis

AML12 cells were seeded in 6-well plates (6 × 10^5^ cells/well) and treated with puerarin or DMSO (control) in DMEM/F12 medium containing 1 mM HFFA for 12 h. Cultured AML12 cells were fixed in buffered formalin. For quantitative analysis of ROS production, cells were incubated with 2’,7’-dichlorodihydrofluorescein diacetate (DCFH-DA, purchased from Sigma-Aldrich; Merck KGaA, Darmstadt, Germany) for 30 min, followed by DAPI staining (purchased from Thermo Fisher Scientific, Waltham, MA, USA) for nuclear staining.

### 4.4. Oil Red O Staining

For the assessment of lipid accumulation in AML12 cells, Oil Red O staining was employed alongside a cellular triglyceride assay. The AML12 cells were cultured in 6-well plates at a density of 6 × 10^5^ cells/well and treated with high concentrations of fatty acids (FFA). Post 48-h treatment, cells were washed thrice with PBS and fixed with 4% formalin at room temperature for 30 min. The cells were gently rinsed with PBS and immersed in 60% isopropanol for 5 min. After isopropanol removal, the cells were stained with Oil Red O solution for 20 min and counterstained with Mayer’s hematoxylin solution for 1 min. Excess dye was eliminated with four washes using distilled water, and the cells were then examined under a bright-field microscope.

### 4.5. Mitochondria Mass and DNA Assay and Mitochondrial Respiratory Complexes

Mitochondria mass was detected according to the method of Kuo et al. [57]. Hepatocytes seeded onto glass coverslips were incubated with 100 nM MitoTracker Red 580 (Invitrogen, Carlsbad, CA, USA). After incubation for 30 min at 37 °C, coverslips were rinsed and washed. DAPI was used for nuclear staining. Finally, the cells were fixed in 4% paraformaldehyde for 15 min and visualized using a microscope.

Crude mitochondrial fraction was extracted from AML12 cells and liver using Mitochondria Isolation Kit (Thermo Fisher Scientific, Waltham, MA, USA) for Tissue followed by manufacturer’s instructions. With the assistance of the Bio-Rad Rapid Coomassie kit, the proteins were measured. For measurement of Mitochondrial Complex I, III, and IV activity in liver, using Mitochondrial Complex activity Assay Kit, respectively [58].

### 4.6. Flow Cytometry for ROS and Mitochondrial Membrane Potential Assay

AML12 cells were treated with 10 μM DCFH-DA (Molecular Probes, Inc., Eugene, OR, USA) and incubated at 37 °C for 40 min prior to the conclusion of the treatment protocol. Subsequently, the cells were washed and gently scraped using ice-cold PBS [57]. The fluorescence of DCF was detected in accordance with the instructions provided by the manufacturer. To assess the mitochondrial membrane potential (MMP), the mean fluorescence intensity of JC-1, using the BD™ MitoScreen kit, was measured. The treated cells were harvested and resuspended at a density of 1 × 10^5^/mL in PBS containing 1 μg/mL JC-1, followed by incubation at 37 °C for 30 min. The samples were then subjected to flow cytometric analysis using a FACSCalibur instrument (BD Biosciences, San Jose, CA, USA) [57]. The results are expressed as the relative percentage of fluorescence.

### 4.7. RNA Isolation and Real-Time PCR Analyses

Total-RNA was extracted from the hepatocytes using the guanidinium-phenol-chloroform method. Total-RNA (5 μg) was reverse-transcribed using the RevertAid™ First Strand cDNA Synthesis kit according to the manufacturer’s protocol. The cDNA was amplified using the TaqDNA polymerase kit (Vilnius, Lithuania). RT-PCR products were separated by electrophoresis on a 3% agarose gel and were quantified by the ImageQuant 5.2 software (Healthcare Bio-Sciences, Philadelphia, PA, USA). Real-time PCR was performed with a LightCycler 1.5 apparatus (Roche Diagnostics, Mannheim, Germany) using the LightCycler FastStart DNA MasterPLUS SYBR-Green I kit according to the manufacturer’s protocol. Mitochondrial DNA copy number was determined by real-time PCR as previously described [57]. Oligonucleotide sequences for-qPCR: *Srebp1*: (F) 5′ actgtcttggttgttgatgagctggagcat 3′, (R) 5′ atcggcgcggaagctgtcggggtagcgtc 3′; *Fas*: (F) 5′ tgtcattggcctcctcaaaaagggcgtcca 3′, (R) 5′ tcaccactgtgggctctgcagagaagcgag 3′; *Scd1*: (F) 5′ ccggagaccccttagatcga 3′, (R) 5′ tagcctgtaaaagatttctgcaaacc 3′; *Acc1*: (F) 5′ cccgccagcttaaggaca 3′, (R)5′ tggatgggatgtgggca 3′; *Tnfα*: (F) 5′ ttgacctcagcgctgagttg 3′, (R) 5′ cctgtagcccacgtcgtagc 3′; *Ifnγ*: (F) 5′ cctcaaacttggcaatactc 3′, (R) 5′ agcaacaacataagcgtcat 3′; *Il-1β*: (F) 5′ aacctgctggtgtgtgacgttc 3′, (R) 5′ cagcacgaggcttttttgttgt 3′; *Mcp1*: (F) 5′ aggtccctgtcatgcttctg 3′, (R) 5′ tctggacccattccttcttg 3′; Nlrp3: (F) 5′ agccttccaggatcctcttc 3′, (R) 5′ cttgggcagcagtttctttc 3′; *Pannexin*: (F) 5′ ggccacggagtatgtgttct 3′, (R) tacagcagcccagcagtatg3′; *Asc*: (F) 5′ gaagctgctgacagtgcaac 3′, (R) 5′ tgtgagctccaagccatacg 3′, *Pro-casp 1*: (F) 5′ agatggcacatttccaggac 3′, (R) 5′ gatcctccagcagcaacttc 3′; *Sirt1*: (F) 5′ gcaacagcatcttgcctgat 3′, (R) 5′ gtgctactggtctcactt 3′; *Pgc1α*: (F) 5′ gactcagtgtcaccaccgaaa 3′, (R) 5′ tgaacgagagcgcatcctt 3′; *Ucp1*: (F) 5′ cctgcctctctcggaaacaa 5′, (R) tgtaggctgcccaatgaaca 3′; *Complex I (20kDa)*: (F) 5′ ccagctgcgcagagttcatc 3′, (R) 5′; gagagagcttggggaccacg 3′; *Complex II (Ip)*: 5′ tctaccgctgccacaccatc 3′, (R) 5′ aagccaatgctcgcttctcc 3′; *Complex III (Core II)*: (F) 5′ ccattggaaatgcagaggca 3′, (R) 5′ ggctggtgacttcctttggc 3′; *Complex IV (Cox2)*: (F) 5′ tcatgagcagtcccctccct 3′, (R) 5′ gccatagaataaccctggtcgg 3′; *Complex V (F1α)*: (F) 5′ atctatgcgggtgtacgggg 3′, (R) 5′ agggactggtgctggctgat 3′; *Pink*: (F) 5′ gagcagactcccagttctcg 3′, (R) 5′ gtcccactccacaaggatgt 3′; *Ndp52*: (F) 5′ aaggactggattggcatcttta 3′, (R) 5′ aggtcagcgtacttgtctttc 3′; *Ambra1*: (F) 5′ gggatgttgtgcctttgca 3′, (R) 5′ cctggtgtgggaagagagaaga 3′; *Phb2*: (F) 5′ cttggttccagtaccccattatc 3′, (R) 5′ cgagacaacactcgcaggg 3′; *Bnip3*: (F) 5′ cagcatgaatctggacgaag 3′, (R) 5′ atcttcctcagacagagtgc 3′; *Nix*: (F) 5′ gagccggatactgtcgtcct 3′, (R) 5′ caatatagatgccgagcccca 3′ Fundc1: (F) 5′ tgtgatatccagcggcttcg 3′, (R) 5′ gccggctgttccttactttg 3′.

### 4.8. Immunofluorescence Staining

AML12 cell (1 × 10^3^ cells) were cultured on glass coverslips. After incubation with HFFA and with or without puerarin, the cells were fixed in 4% paraformaldehyde for 15 min, washed with ice-cold PBS and blocked with 7.5% normal goat serum for 30 min at room temperature. After washing with ice-cold PBS twice, the cells were incubated with anti-SREBP1, anti-NF-κB, anti-FXR, anti-BESP, and anti-CYP7A1 anti-bodies for 1 h at room temperature. After washes with ice-cold PBS twice, the cells were incubated with diluted FITC-conjugated secondary anti-body for 1 h at room temperature. In addition, DAPI was used for nuclear staining. The slides were mounted in mounting medium and visualized using a microscope (Olympus, Tokyo, Japan). Anti-bodies used for immunohistochemistry and immunofluorescence: SREBP1 (ab28481, abcam, Cambridge, UK); NF-κB (p65, sc-8008, Santa Cruz Biotechnology, Dallas, TX, USA); FXR (ab51970, abcam, Cambridge, UK); BSEP (ab217532, abcam, Cambridge, UK); CYP7A1 (BS-2399R, Bioss, Woburn, MA, USA); CD36 (SC-70644, Santa Cruz Biotechnology, Dallas, TX, USA); MCP-1 (ab7202, abcam, Cambridge, UK); NLRP (ab214185, abcam, Cambridge, UK); SIRT1 (ab110304, abcam, Cambridge, UK); PGC1α (ab54481, abcam, Cambridge, UK); COXIII (ab14745, abcam, Cambridge, UK); DAPI (ab104139, abcam, Cambridge, UK); UCP1 (ab10983, abcam, Cambridge, UK); COXIV (ab14705, abcam, Cambridge, UK); Parkin (GTX39745, GeneTex, Irvine, CA, USA); Ubiquitin (ab7254, abcam, Cambridge, UK); Cyt C (ab13575, abcam, Cambridge, UK); pp62 (Ser403, GTX128171, GeneTex, Irvine, CA, USA); NTCP (ab131084, abcam, Cambridge, UK).

### 4.9. Experimental Animals

Male C57BL/6 mice were acquired from the National Animal Center and utilized in experiments upon reaching a weight of 25–28 g. An obese animal model was established via a 16-week high-fat diet regimen [59]. FXR knockout (KO) mice were sourced from the Jackson Laboratory (stock no. 007214). All mice were kept under controlled conditions, including a 24 °C environment with a 12–12 h light-dark cycle, and had ad libitum access to food and water at the Chang Gung University Laboratory Animal Center. A total of 30 mice were divided into 6 groups: (1) normal mice + vehicle solution (mixing 33% ethanol, 33% DMSO, and 33% Tween-80, then diluting this 1:50 in 0.5 M NaCl) for 4 weeks; (2) HFD-induced obese mice fed a Rodent Diet with 40% of calories from fat (mostly palm oil), 20% from fructose, and 2% cholesterol for 16 weeks (research diets, catalog number D09100310) + vehicle solution for 4 weeks; (3) obese mice given puerarin (100 mg/kg orally once daily) for 4 weeks; (4) FXR KO mice + vehicle solution for 4 weeks; (5) HFD-fed FXR KO mice + vehicle solution for 4 weeks; (6) FXR KO mice fed both HFD and puerarin (100 mg/kg orally once daily) for 4 weeks. The puerarin stock solution was prepared by mixing 33% ethanol, 33% DMSO, and 33% Tween-80, then diluting this 1:50 in 0.5 M NaCl, utilizing puerarin from Sigma-Aldrich (cat no. 82435). The vehicle solution and puerarin were administered to the mice during the last 4 weeks of the experiment.

### 4.10. Histopathology and Immunohistochemistry

Following euthanasia, liver tissues from experimental mice were fixed in 10% formalin and embedded in paraffin. Tissue sections of 5 μm thickness were stained with hematoxylin and eosin for histological analysis. Additional immunohistochemical methods were applied with scoring based on a standardized histopathology score for hepatic lesions [20]. ImageJ (Version 1.54i) was utilized for analysis post-IHC staining, comparing other groups to the control group to accurately quantify and compare protein expression levels under different conditions. This comprehensive approach ensures detailed examination and comparison of tissue pathology.

### 4.11. Gut Microbiota Analysis via 16S rRNA and Intestinal Tissue DNA Extraction

Upon completion of the experimental procedures, fecal samples were harvested from the large intestines of mice and subsequently stored at −80 °C for preservation. The extraction of DNA from these fecal samples was carried out using the QIAGEN mini stool kit (QIAGEN, Valencia, CA, USA). For this process, a 200 mg fecal sample was placed into a 2 mL centrifuge tube, to which 1 mL of InhibitEX Buffer was added. The mixture was then incubated at room temperature to ensure proper integration of the components. Following thorough mixing, the sample was subjected to a heat treatment at 70 °C for 5 min and agitated for 15 s to ensure uniformity. The sample was then centrifuged at a force of 20,000× *g* for 1 min, facilitating the precipitation of fecal particles. Subsequently, 200 μL of the supernatant was carefully transferred into a new 1.5 mL centrifuge tube. To this tube, 15 μL of proteinase K and 200 μL of Buffer AL were added, and the mixture was again heated at 70 °C for 10 min. A brief centrifugation followed to eliminate any droplets from the cap of the tube. The next step involved the addition of 200 μL of absolute ethanol to the mixture, which was then thoroughly mixed. The lysate was then meticulously transferred to a QIAamp spin column and centrifuged at 20,000× *g* for 1 min, after which the filtrate was discarded. The column underwent a washing step with 500 μL of Buffer AW1, followed by centrifugation at 20,000× *g* for 1 min. The column was then placed into a new 2 mL collection tube, and the filtrate was again discarded. Upon opening the column, 500 μL of Buffer AW2 was added, and the column was centrifuged at 20,000× *g* for 3 min. The filtrate was discarded, and the column was transferred to a new 2 mL collection tube and centrifuged at full speed for an additional 3 min. Finally, the spin column was placed into a newly labeled 1.5 mL microcentrifuge tube, and 200 μL of Buffer ATE was directly applied onto the QIAamp membrane. The tube was then left to incubate at room temperature for 1 min before being centrifuged at 20,000× *g* for 1 min, resulting in the collection of the DNA sample.

### 4.12. 16S rRNA Analysis

The 16S Ribosomal DNA sequence is characterized by 9 variable regions (V1-V9), which are instrumental in species differentiation. Gut microbiota analysis predominantly targets the V3-V4 regions, employing primers V3F (5′ CCTACGGGNGGCWGCAG 3′) and V4R (5′ GACTACHVGGGTATCTAATCC 3′). This is followed by Next Generation Sequencing (NGS) for detailed examination. The obtained 16S rDNA sequences are then cross-referenced with the National Center for Biotechnology Information (NCBI) nucleotide database and the PhiX control library for comparative analysis. Operational taxonomic units (OTUs) that exhibit a 97% identity threshold are aggregated into a single OTU using the UPARSE algorithm [60]. For the construction of the taxonomy composition distribution, sequencing reads are meticulously aligned with an array of 16S rRNA sequence databases utilizing Bowtie2 [61]. The PCA plot, which graphically represents the variances and proximities correlating to the interrelationship between two mice, is generated using ClustVis, an online tool developed by the University of Oxford [62]. This PCA plot serves as a visual aid to elucidate the comparative microbial profiles of the subjects under study.

### 4.13. Statistical Analysis

Data are presented as the mean ± SEM. The statistical analyses were performed using a one-way analysis of variance followed by the Student Newman-Keuls multiple-range test. *p* < 0.05 denote statically significant differences.

## 5. Conclusions

Puerarin is effective in ameliorating obesity-associated lipid metabolism disorders and gut microbiota dysbiosis, largely by activating hepatic FXR signaling and restoring FXR-mediated homeostasis. The beneficial metabolic effects of puerarin require functional FXR activity in adipose and liver tissues, as evidenced by reduced efficacy in FXR knockout mice. Puerarin’s therapeutic impact also hinges on its ability to enhance mitophagy, which plays a critical role in maintaining mitochondrial and overall cellular health in the management of obesity.

## Figures and Tables

**Figure 1 ijms-25-05274-f001:**
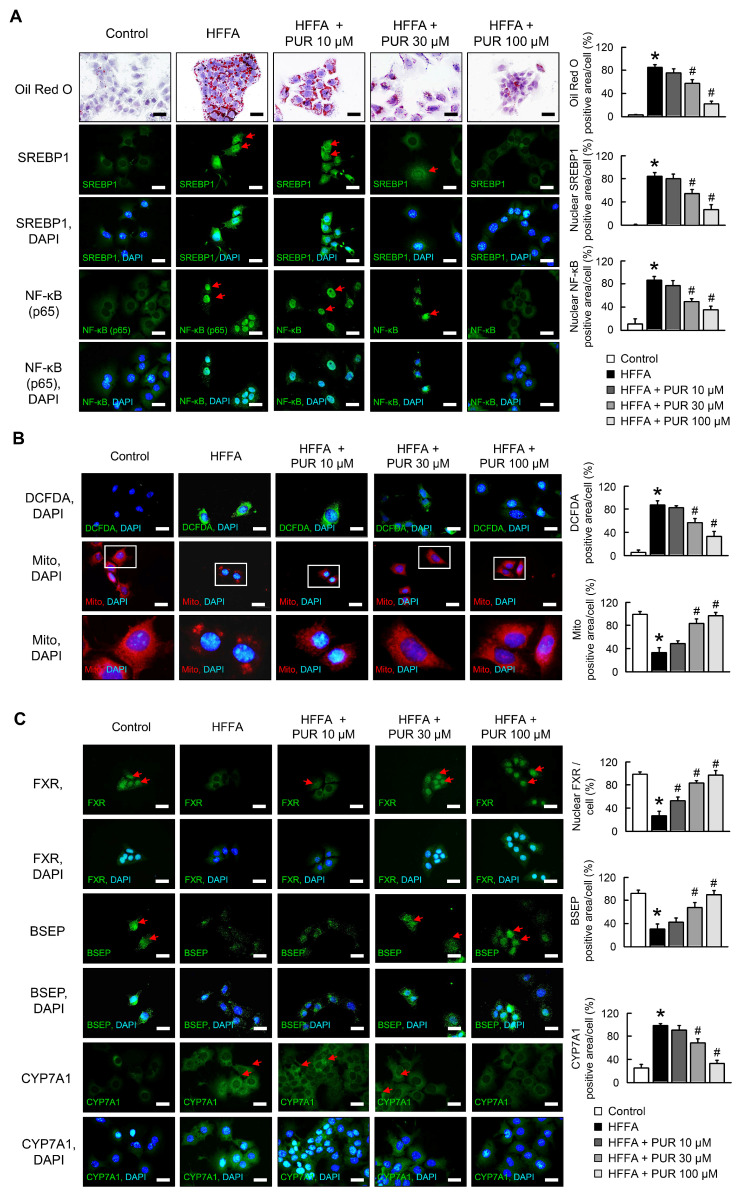
Effects of puerarin on HFFA-mediated lipid accumulation, NF-κB nuclear translocation, mitochondrial function, and FXR signaling in AML12 cells. (**A**) Oil-Red O (red) staining reveals intracellular lipid accumulation, and immunofluorescence indicates SREBP1 (green) and NF-κB (p65, green) levels. (**B**) Visualization of reactive oxygen species (ROS, green) and mitochondrial integrity. (**C**) Expression levels of FXR (green), BSEP (green), and CYP7A1 (green) demonstrated through immunofluorescence. Scale bars, 20 µm. Nuclei of corresponding cells visualized by DAPI staining. Red arrow highlights positive staining. The white box in the image indicates the region of positive area, which is magnified and displayed below for detailed examination. * *p* < 0.05, control vs. HFFA; # *p* < 0.05, HFFA vs. HFFA + PUR.

**Figure 2 ijms-25-05274-f002:**
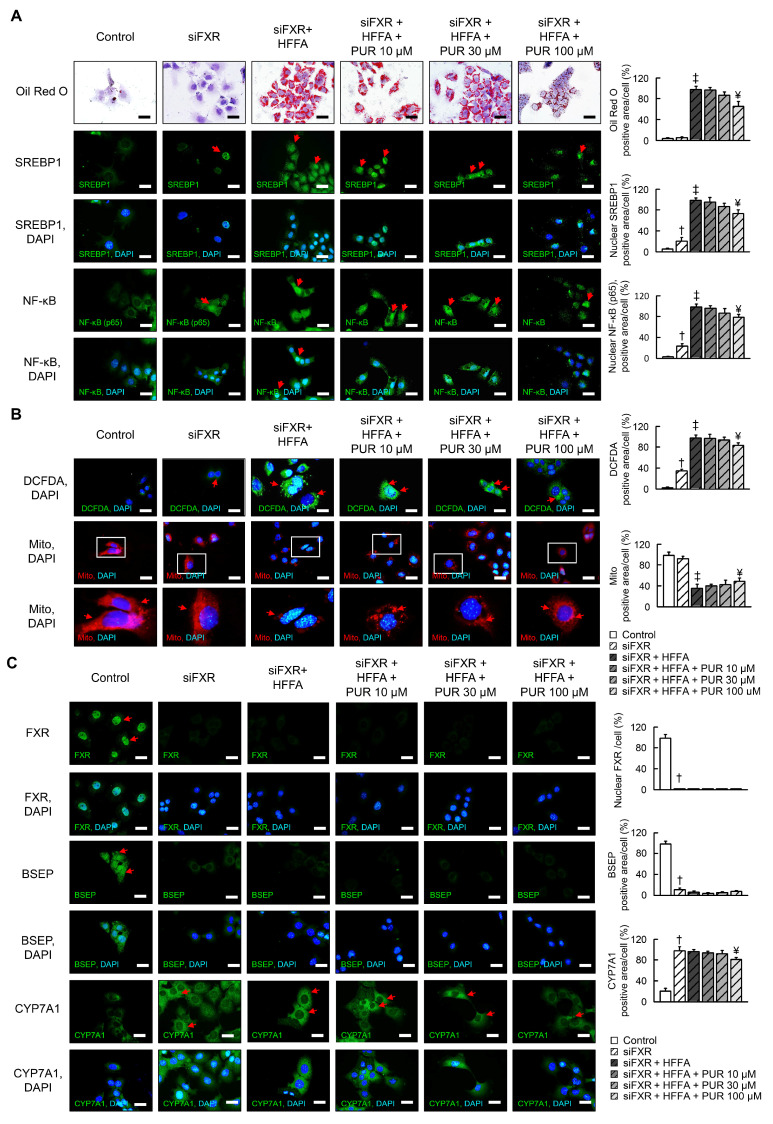
Impact of puerarin on cellular responses in AML12 cells treated with siFXR and HFFA. (**A**) Assessment of intracellular lipid inclusions and localization of SREBP1 (green) and NF-κB (p65, green) proteins using fluorescence. (**B**) Visualization of reactive oxygen species (ROS, green) and mitochondrial integrity. (**C**) Expression of FXR (green), BSEP (green), and CYP7A1 (green) indicated through immunofluorescence. Nuclei of corresponding cells visualized by DAPI (blue)staining. Scale bars, 20 µm. Red arrow highlights positive staining. The white box in the image indicates the region of positive area, which is magnified and displayed below for detailed examination. † *p* < 0.05, control vs. siFXR; ‡ *p* < 0.05, siFXR vs. siFXR + HFFA; ¥ *p* < 0.05, siFXR + HFFA vs. siFXR + HFFA + PUR.

**Figure 3 ijms-25-05274-f003:**
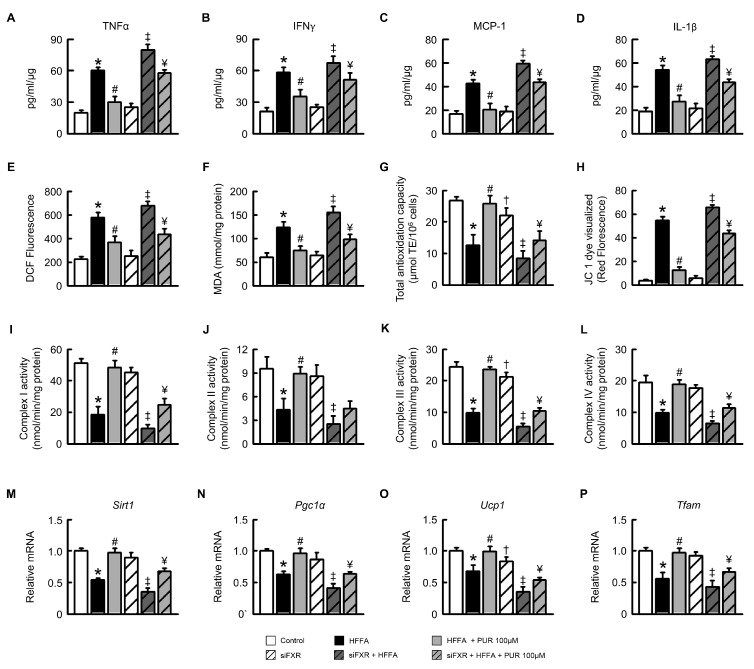
Puerarin mitigates HFFA-induced inflammatory factors, ROS, and mitochondrial biogenesis and activity impairment in AML12 cells. (**A**–**D**) Concentrations of pro-inflammatory cytokines TNFα, IFNγ, MCP-1, and IL-1β. (**E**,**F**) Analysis of ROS and MDA levels. (**G**) Assessment of anti-oxidant capacity. (**H**) Analysis of mitochondrial damage. (**I**–**L**) Enzymatic activities of Complexes I, II, III, and IV. (**M**–**P**) Expression levels of *Sirt1*, *Pgc1α*, *Ucp1*, and *Tfam* mRNA. Statistical significance denoted as follows: * *p* < 0.05, control vs. HFFA; # *p* < 0.05, HFFA vs. HFFA + PUR; † *p* < 0.05, control vs. siFXR; ‡ *p* < 0.05, siFXR vs. siFXR + HFFA; ¥ *p* < 0.05, siFXR+ HFFA vs. siFXR + HFFA + PUR.

**Figure 4 ijms-25-05274-f004:**
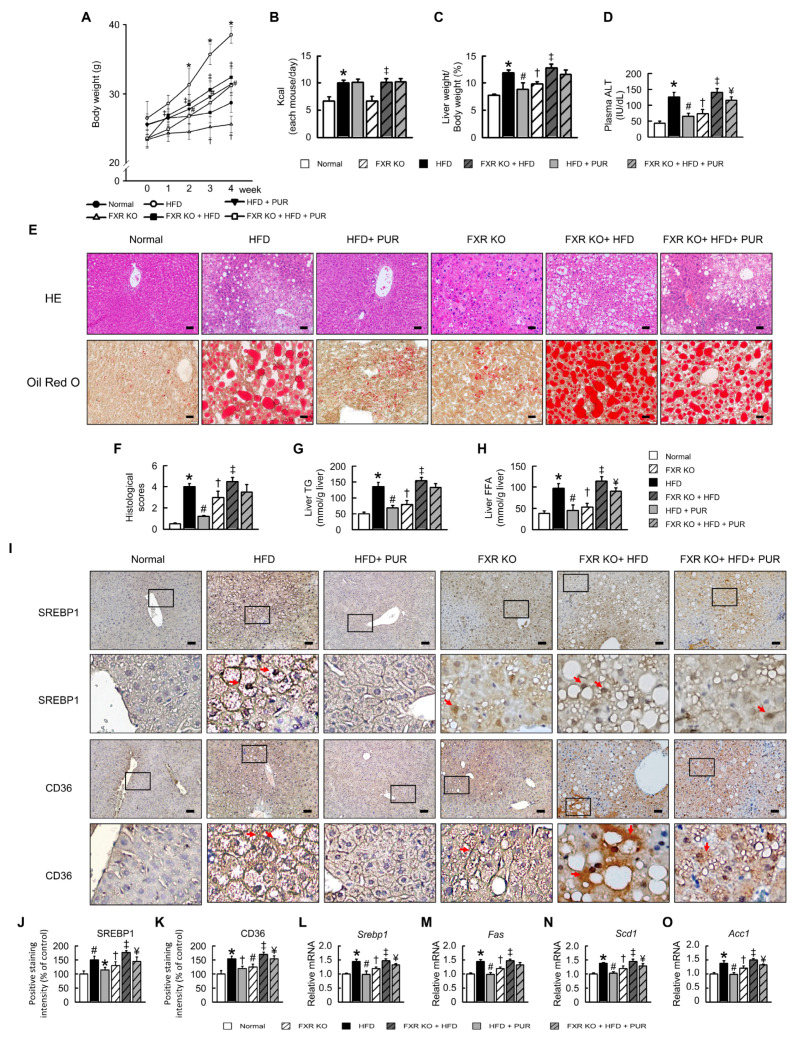
Effects of puerarin on liver damage and lipid metabolism in HFD-induced obese mice. (**A**) Body weight (g). (**B**) Food intake (Kcal). (**C**) Liver weight/Body weight (%). (**D**) Plasma ALT. (**E**) Representative pictures of HE and Oil Red O staining of liver section. (**F**) Histological scores. (**G**) Liver TG. (**H**) Liver FFA. (**I**) Immunohistochemical staining of CD36 and SREBP-1c in liver. Red arrow highlights positive staining. Scale bars, 50 µm. Red arrow highlights positive staining. (**J**,**K**) Positive staining intensity of SREBP1 and CD36. (**L**–**O**) qRT-PCR analysis of *Srebp1*, *Fas*, *Scd1*, and *Acc1* mRNA expression in liver. Relative mRNA expression normalized to *Gapdh* and controls. The black box in the image indicates the region of positive area, which is magnified and displayed below for detailed examination. In all panels, results are expressed as mean ± S.E.M. of five independent experiments. * *p* < 0.05, normal vs. HFD; # *p* < 0.05, HFD vs. HFD + PUR; † *p* < 0.05, normal vs. FXR KO; ‡ *p* < 0.05, FXR KO vs. FXR KO + HFD; ¥ *p* < 0.05, FXR KO + HFD vs. FXR KO + HFD + PUR.

**Figure 5 ijms-25-05274-f005:**
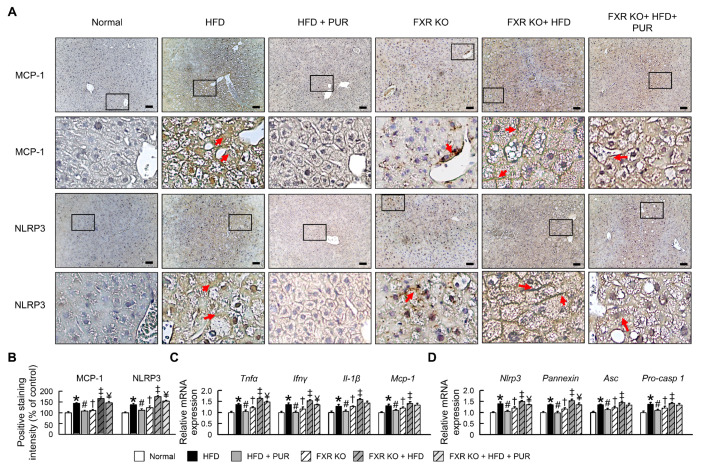
Puerarin attenuates inflammatory response in HFD-induced obese mice. (**A**) Protein expression levels of NLRP3 and MCP-1 in liver. Red arrow highlights positive staining. Scale bar: 50 μm. The black box in the image indicates the region of positive area, which is magnified and displayed below for detailed examination. (**B**) Positive staining intensity of NLRP3 and MCP-1. Red arrow highlights positive staining. Expression levels of inflammatory cytokines IL-1β, in liver. mRNA expression (**C**) *Tnfα*, *Ifnγ*, *Il-1β*, *Mcp-1*, (**D**) inflammasome-related factors *Nlrp3*, *Pannexin*, *Asc*, and *Pro-casp 1* in liver. * *p* < 0.05, normal vs. HFD; # *p* < 0.05, HFD vs. HFD + PUR; † *p* < 0.05, normal vs. FXR KO; ‡ *p* < 0.05, FXR KO vs. FXR KO + HFD; ¥ *p* < 0.05, FXR KO + HFD vs. FXR KO + HFD + PUR.

**Figure 6 ijms-25-05274-f006:**
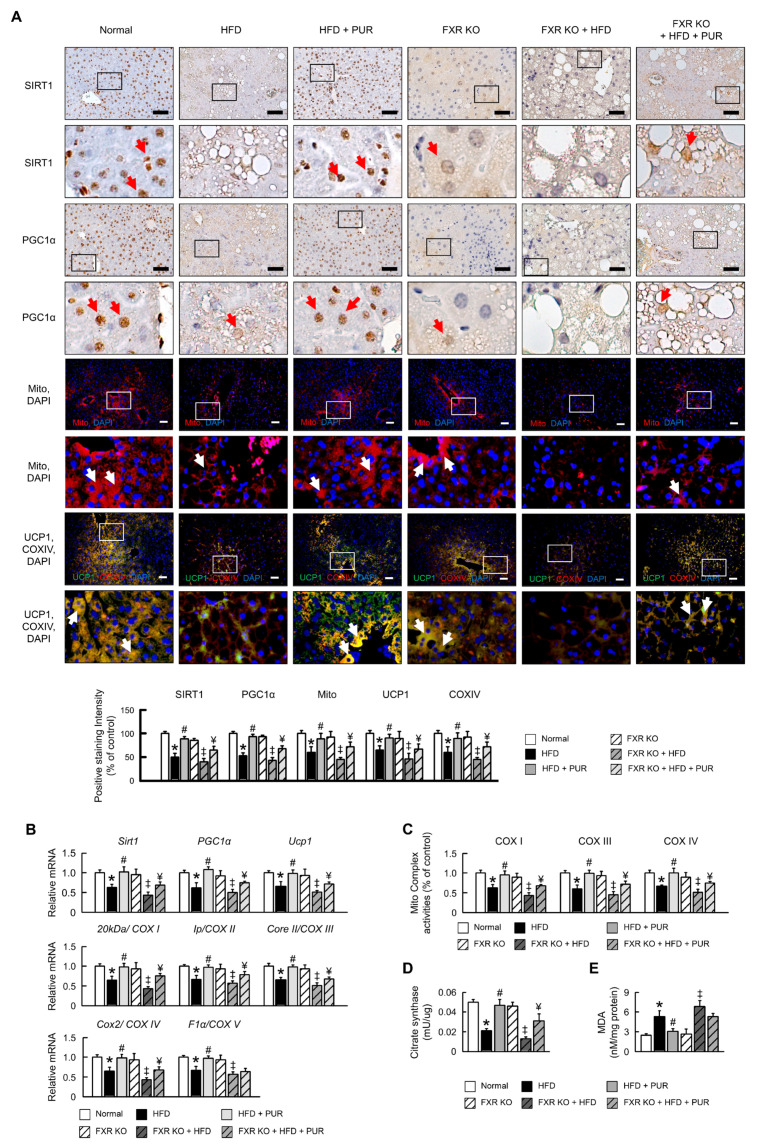
Regulation of mitochondrial biogenesis by puerarin in HFD-induced obesity mice. (**A**) Protein expression levels of SIRT1 and PGC1α. Immunofluorescence images showing protein expression levels of mitochondria (red), mitochondrial complex IV (red), and UCP1 (green) in liver. Scale bars, 50 µm. White arrow highlights positive staining. Red and white arrow highlights positive staining. qRT-PCR analysis of (**B**) *Sirt1*, *Pgc1α*, *Ucp1*, *COXI*, *COX II*, *COXVI*, and *COXV* mRNA expression in liver. Relative mRNA expression was normalized to *Gapdh* and then normalized to controls. (**C**) Enzymatic activities of Complexes I, III, and IV. (**D**,**E**) Analysis of Citrate synthase and MDA levels. The black and white box in the image indicates the region of positive area, which is magnified and displayed below for detailed examination. In all panels, results are expressed as mean ± S.E.M. of five independent experiments * *p* < 0.05, normal vs. HFD; # *p* < 0.05, HFD vs. HFD + PUR; ‡ *p* < 0.05, FXR KO vs. FXR KO + HFD; ¥ *p* < 0.05, FXR KO + HFD vs. FXR KO + HFD + PUR.

**Figure 7 ijms-25-05274-f007:**
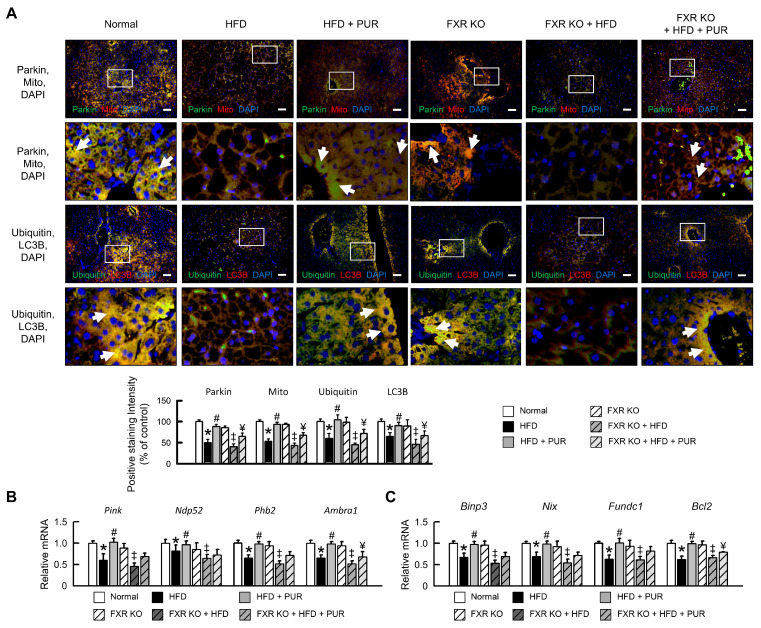
Regulation of mitophagy by puerarin in HFD-induced obesity mice. (**A**) Immunofluorescence images showing protein expression levels of Parkin, mitochondria, ubiquitin, and LC3B in liver. White arrow highlights positive staining. Scale bar: 50 μm. qRT-PCR analysis of (**B**) *Pink*, *Ndp52*, *Phb2*, *Ambra1*, (**C**) *Binp3*, *Nix*, *Fundc*, and *Bcl2* mRNA expression in liver. Relative mRNA expression normalized to *Gapdh* and then normalized to controls. The white box in the image indicates the region of positive area, which is magnified and displayed below for detailed examination. In all panels, results are expressed as mean ± S.E.M. of five independent experiments. * *p* < 0.05, normal vs. HFD; # *p* < 0.05, HFD vs. HFD + PUR; ‡ *p* < 0.05, FXR KO vs. FXR KO + HFD; ¥ *p* < 0.05, FXR KO + HFD vs. FXR KO + HFD + PUR.

**Figure 8 ijms-25-05274-f008:**
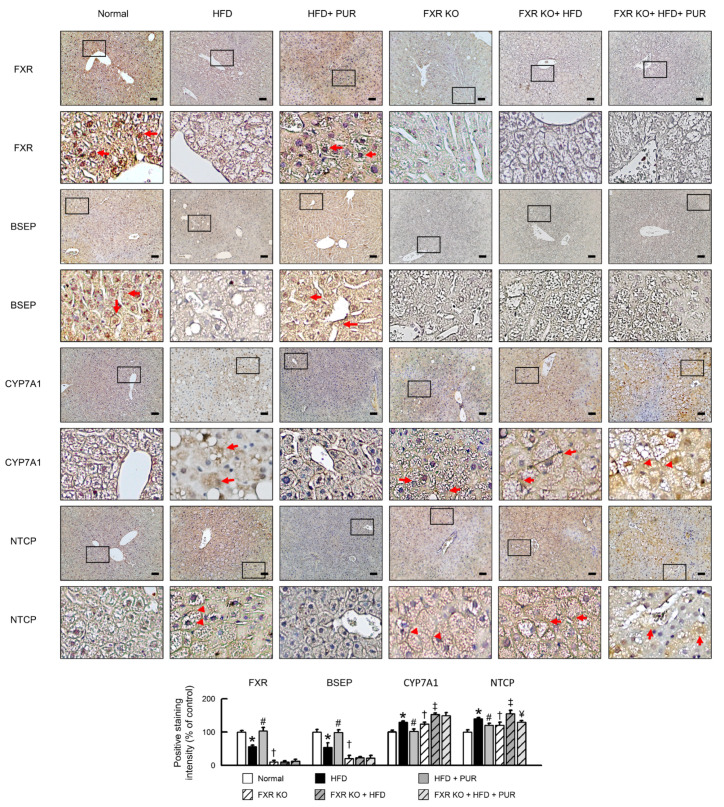
Impact of puerarin on bile acid transport proteins and FXR signaling in obese mice. Representative FXR, BSEP, CYP7A1, and NTCP staining of liver. Scale bar: 50 μm. Red arrow highlights positive staining. Quantification of FXR, BSEP, CYP7A1, and NTCP protein levels by positive staining of liver. The black box in the image indicates the region of positive area, which is magnified and displayed below for detailed examination. In all panels, results are expressed as mean ± S.E.M. of five independent experiments. * *p* < 0.05, normal vs. HFD; # *p* < 0.05, HFD vs. HFD + PUR; † *p* < 0.05, normal vs. FXR KO; ‡ *p* < 0.05, FXR KO vs. FXR KO + HFD; ¥ *p* < 0.05, FXR KO + HFD vs. FXR KO + HFD + PUR.

**Figure 9 ijms-25-05274-f009:**
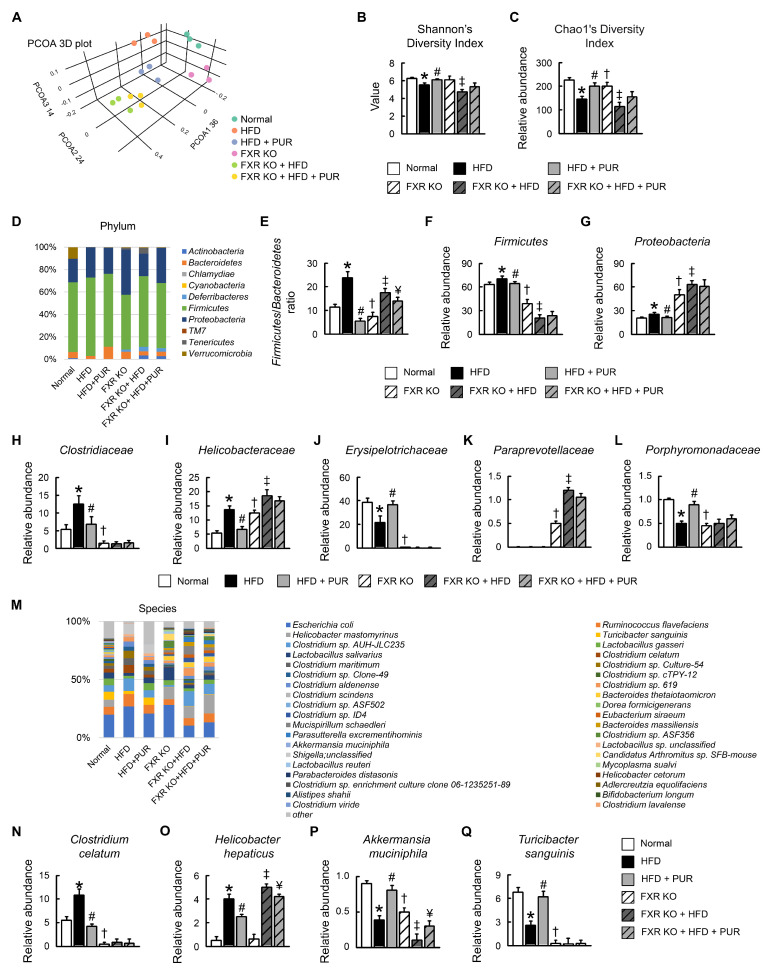
Puerarin alters intestinal microbial composition in mice. (**A**) PCA plot based on abundance of bacterial gene sequences in fecal content. Axes correspond to principal component 1 (x axis) and 2 (y axis). Alpha diversity measurements of microbiota across locations. (**B**) Shannon and (**C**) Chao1′s diversity index. (**D**) Microbial community bar plot by phylum relative abundance (%). (**E**) *Firmicutes/Bacteroidetes* ratio. Abundance of (**F**) *Firmicutes* and (**G**) *Proteobacteria* by phylum. Abundance of (**H**–**L**) *Clostridiaceae*, *Helicobacteraceae*, *Erysipelotrichaceae*, *Paraprevotellaceae*, and *Porphyromonadaceae* by family. (**M**) Microbial community bar plot by species relative abundance (%). Abundance of (**N**) *Clostridium celatum*, (**O**) *Helicobacter hepaticus*, (**P**) *Akkermansia muciniphila*, and (**Q**) *Turicibacter sanguinis* by species. In all panels, results are expressed as mean ± S.E.M. of five independent experiments. * *p* < 0.05, normal vs. HFD; # *p* < 0.05, HFD vs. HFD + PUR; † *p* < 0.05, normal vs. FXR KO; ‡ *p* < 0.05, FXR KO vs. FXR KO + HFD; ¥ *p* < 0.05, FXR KO + HFD vs. FXR KO + HFD + PUR.

**Table 1 ijms-25-05274-t001:** Histopathology score of hepatic lesions.

Histological	Severity	Description	Score
Steatosis	Absent	<10%	0
	Mild	10–30%	1
	Marked	31–60%	2
	Severe	>60%	3
Inflammation	None		0
	Moderate	Scattered ^a^	1
	Marked	Focia ^a^	2
	Severe	Diffuse ^a^	3
Necrosis	Absent	0%	0
	Mild	<10%	1
	Marked	10–30%	2
	Severe	>50%	3
Fibrosis ^b^	Absent		0
	Mild		1
	Marked		2
	Severe		3

^a^ Amount of inflammatory cells; ^b^ Mild = moderately thickened centrolobular vein (CLV), marked = markedly thickened CLV, severe = cirrhosis [20].

## Data Availability

Data is contained within the article.

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
