# Peer review of "Puerarin Modulates Hepatic Farnesoid X Receptor and Gut Microbiota in High-Fat Diet-Induced Obese Mice"

_ijms, 2024, doi:10.3390/ijms25105274_

Round 1
Reviewer 1 Report (Previous Reviewer 1)
Comments and Suggestions for Authors
The authors have addressed most of my critiques.
Minor Comments
1) Line 225 states "ration" - this should be "ratio".
2) Line 604 - a reference is missing.
Comments on the Quality of English LanguageN/A
Author Response
Reviewer 1 comments
The authors have addressed most of my critiques.
Minor Comments
1) Line 225 states "ration" - this should be "ratio".
2) Line 604 - a reference is missing.
Response
- The incorrect has revised. The revised text is marked in red with an underline. (Please see line 225)
- A reference has been added to support the information provided. (Please see line 603)

Reviewer 2 Report (Previous Reviewer 2)
Comments and Suggestions for Authors
In this study, Yang et al. seek the effects of puerarin in obesity using high fat diet models. The authors previously submitted this manuscript (ijms-2851933), and there were multiple flaws and issues to be fixed. The authors improved images and added new data to polish the study and address my previous comments. Data look solid and I have no major criticisms for this study. One minor suggestion is that the authors say as if puerarin improves liver conditions by increasing gut microbiota diversity, but this study does not prove that. Figure 9 shows that puerarin improves diversity, but it is unknown this change promotes expression of FXR or inhibition of steatosis. The authors should elaborate the manuscript because some descriptions are misleading. Gut microbiota was altered by puerarin, that is all the authors find. This may or may not be associated with FXR expression and steatosis. The authors can discuss about this but should not exaggerate that puerarin inhibits steatosis by improving gut bacteria diversity.
Author Response
Reviewer 2 comments
In this study, Yang et al. seek the effects of puerarin in obesity using high fat diet models. The authors previously submitted this manuscript (ijms-2851933), and there were multiple flaws and issues to be fixed. The authors improved images and added new data to polish the study and address my previous comments. Data look solid and I have no major criticisms for this study. One minor suggestion is that the authors say as if puerarin improves liver conditions by increasing gut microbiota diversity, but this study does not prove that. Figure 9 shows that puerarin improves diversity, but it is unknown this change promotes expression of FXR or inhibition of steatosis. The authors should elaborate the manuscript because some descriptions are misleading. Gut microbiota was altered by puerarin, that is all the authors find. This may or may not be associated with FXR expression and steatosis. The authors can discuss about this but should not exaggerate that puerarin inhibits steatosis by improving gut bacteria diversity.
Response:
We acknowledge the reviewer's valid point. Our study does not definitively establish a causal relationship between the observed changes in gut microbiota diversity induced by puerarin treatment, the increased expression of FXR, and the inhibition of steatosis. We recognize this as a limitation of our work.
We have made some limitations in discussion section:
“Our study provides valuable insights into the association among puerarin treatment, altered gut microbiota composition and diversity, increased FXR expression, and reduced steatosis. However, it is crucial to acknowledge some limitations. While we observed these associations, our study does not conclusively establish a causal relationship between the puerarin-induced changes in gut microbiota diversity, FXR expression, and steatosis inhibition. The precise mechanisms underlying these observations warrant further investigation. We hypothesize that the observed effects may be mediated through a similar mechanism, but further mechanistic studies are necessary to delineate the precise pathways involved. Additionally, our study focused specifically on the potential beneficial effects of puerarin in combating obesity by promoting favorable changes in the gut microbiota. However, the broader implications and applications of these findings, such as the role of puerarin in other metabolic disorders or the potential clinical applications, require further exploration and research.” (line 616-628)

This manuscript is a resubmission of an earlier submission. The following is a list of the peer review reports and author responses from that submission.
Round 1
Reviewer 1 Report
Comments and Suggestions for Authors
Yang et al investigated the effects of the isoflavone puerarin on hepatic lipid and mitochondrial metabolism. The authors found that puerarin may be acting through FXR. However, more experimental data are required to support the conclusions as presented.
Major Comments
1. Overall, many of the methods should be described in more detail.
a. Presumably, puerarin was administered to mice via oral gavage but this is not specified. Were the doses administered daily or by a different schedule?
b. What was the vehicle? Is it soluble in aqueous solution? If a vehicle like corn oil or carboxymethylcellulose was used, this may influence caloric intake or the gut microbiome compared to control.
c. The histology scoring methods should be described in more detail.
d. IHC quantification of proteins described in lines 743-744 for Figs 4E, 5A, 6A, 7A, and 8 is inadequate.
2. Lines 810-812 state that one-way ANOVA was performed for statistical analysis, but the legends for Figs 4, 6-10 state that Student's t-tests were used. T-tests are only to be used when testing 2 groups; performing multiple t-tests greatly increases the risk of Type I errors. All datasets should be re-analyzed using ANOVA and the appropriate post-hoc analyses.
3. Food intake should be shown for all groups, either in grams or kcal.
4. Overall, the histology and IHC figures may need to be presented differently. As it stands, the red arrows indicating positive staining are indistinguishable from the surrounding tissue even when zoomed in to 200%, particularly for CD36, NLRP3, BSEP, and CYP7A1.
5. Similarly, the mitochondrial staining in Figs 6 and 7 may need supplementation. SIRT1 is a deacetylase with many functions, and circulates between nucleus and cytoplasm, while PGC1a is a ubiquitous transcriptional regulator with many functions beyond mitochondrial regulation. There is no indication that the staining depicted in Figs 6 and 7 are localized to the mitochondria. Western blotting of protein isolated from mitochondria would greatly support these conclusions.
6. The bile acid data in Fig 9 is difficult to interpret. The methods state the bile acids were quantified in relation to the internal standard, thus the relative abundance can only be compared within each bile acid. However, bile acid composition is regulated by CYP7A1, CYP8B1, conjugation, the gut microbiome, portal recirculation from intestine to liver, and hepatic reabsorption. It is also not clear how the individual bile acids measured were selected. T-conjugated MCA and CA comprise about 90% of bile acids in mice (PMID: 18801708), and TCA is not presented. Glycine-conjugated bile acids like GDCA are rare in mice and may not hold much physiological significance. If possible, these data should be quantified in nmol or umol.
7. Fig 10 depicts populations of the gut microbiota. The Y-axis is presented as relative abundance, which typically implies % of total counts detected. This is unclear, as Normal mice across the graphs add up to more than 100%. This figure needs to be clarified (is this instead depicting raw bacterial counts detected? What is the data relative to?).
8. It's not entirely clear how the microbiome data supports the rest of the paper. Phylum-level changes do not provide much information related to changes in bile acids or steatotic status other than F:B ratio. If possible, deeper sequencing information should be provided.
9. Overall, the conclusion can be improved. There is detailed discussion about changes in primary bile acids vs secondary bile acids without explanation of why certain bile acids were omitted from analysis (primary TCA and CDCA/TCDCA, secondary LCA and TLCA). Lines 546 and 556 incorrectly states that DCA is a primary bile acid.
10. Line 566 states that puerarin increases bile acid excretion from the liver - this can be support by measuring bile acids in gallbladder, intestinal tissue, and feces.
Minor Comments
1. The legend in Fig 3 does not match the order presented in the graph - this can be re-arranged to make the figure easier to interpret.
2. Line 739 states that intestinal tissue was harvested for staining but this data was not presented.
Comments on the Quality of English LanguageMinor errors that can be easily corrected.
Author Response
Response to Reviewer 1
Major Comments
- Overall, many of the methods should be described in more
- Presumably, puerarin was administered to mice via oral gavage but this is not Were the doses administered daily or by a different schedule?
- What was the vehicle? Is it soluble in aqueous solution? If a vehicle like corn oil or carboxymethylcellulose was used, this may influence caloric intake or the gut microbiome compared to
- The histology scoring methods should be described in more
- IHC quantification of proteins described in lines 743-744 for Figs 4E, 5A, 6A, 7A, and 8 is
Reply
Thank you for your insightful comments and suggestions regarding the methodological details in our manuscript. We appreciate the opportunity to provide more clarity and address your concerns. Please find our point-by-point responses below:
1a. Puerarin Administration: We acknowledge the lack of clarity regarding the administration of puerarin to the mice. In the revised manuscript, we have specified that puerarin was administered via oral gavage on a daily basis. (Methods Section, Page 24, Line 728)
1b. Vehicle Composition and Potential Effects: Puerarin was dissolved in a mixture of 33% ethanol, 33% DMSO, and 33% Tween-80, diluted 1:50 in 0.5 M NaCl. This composition was chosen based on its ability to effectively dissolve puerarin, as recommended by the manufacturer (Sigma-Aldrich, cat no. 82435). (Methods Section, Page 24, Line 729-731)
We understand your concern regarding the potential impacts of the vehicle on caloric intake and gut microbiome composition. To ensure experimental consistency, the control group mice were also treated with the same vehicle. Our observations revealed no significant differences in caloric intake between the treated and control groups, suggesting that the vehicle likely does not have a significant effect on caloric intake or gut microbiome composition.
1c. Histology Scoring Methods: We have enhanced the description of our histology scoring methods in the revised manuscript. Scoring was based on a standardized histopathology score for hepatic lesions, as detailed in Table 1. This standardization ensures consistent and reproducible assessment across all samples. (Methods Section, Page 24, Line 735-737, and Table I)
1d. Immunohistochemistry (IHC) Quantification: For quantification of protein expression levels in Figures 4E, 5A, 6A, 7A, and 8, we utilized ImageJ software, which enables precise and objective comparison among groups. We have clarified this in the Methods section of the revised manuscript. (Methods Section, Page 24, Line 737-739)
We hope that these revisions and clarifications have addressed your concerns regarding the methodological details in our study.
- Lines 810-812 state that one-way ANOVA was performed for statistical analysis, but the legends for Figs 4, 6-10 state that Student's t-tests were T-tests are only to be used when testing 2 groups; performing multiple t-tests greatly increases the risk of Type I errors. All datasets should be re-analyzed using ANOVA and the appropriate post-hoc analyses.
Reply
We appreciate you bringing this to our attention, as it has allowed us to thoroughly re-evaluate our analyses and ensure the appropriate statistical tests were applied.
Regarding the statistical analysis methods:
(1) Analysis for Datasets with More than Two Groups: We confirm that one-way ANOVA was performed for all datasets involving more than two groups, followed by the Student Newman-Keuls multiple-range test to identify significant differences between specific groups. To ensure clarity and consistency, we have revised the statistical analysis section in the Methods (Page 24, Lines 784-786)
(2) Correction in Figure Legends: We acknowledge the incorrect statement of using Student's t-tests in the legends for Figures 4, 6-10. We have updated the figure legends to accurately reflect the use of one-way ANOVA analyses for these datasets.
- Food intake should be shown for all groups, either in grams or
Reply
To address your comment, we have added the food intake data to Figure 4B in the revised manuscript. The food consumption for each mouse group is now presented in kilocalories (Page 9). By providing this additional information, readers will have a comprehensive understanding of the dietary intake across all experimental conditions, allowing for a more informed interpretation of the observed effects and outcomes.
- Overall, the histologyand IHC figures may need to be presented As it stands, the red arrows indicating positive staining are indistinguishable from the surrounding tissue even when zoomed in to 200%, particularly for CD36, NLRP3, BSEP, and CYP7A1.
Reply
Thank you for highlighting the issue regarding the visibility of the histology and immunohistochemistry (IHC) figures, particularly the red arrows indicating positive staining. We appreciate you taking the time to thoroughly examine these figures and provide these valuable opinions
To address your concern, we have taken the following steps to improve the clarity and presentation of these images:
(1) Increased Resolution: We have increased the resolution of all histology and IHC images, ensuring that the red arrows indicating positive staining are clearly distinguishable from the surrounding tissue, even when zoomed in.
(2) Enlarged Red Arrows: To further enhance visibility, we have enlarged the red arrows, providing a better view field and making it easier to identify the regions of positive staining.
(3) Higher Magnification Insets: Each figure now includes higher magnification insets that specifically highlight areas of positive staining for markers such as CD36, NLRP3, BSEP, and CYP7A1. These insets provide a clear view of the specific staining patterns mentioned in the text, allowing for better visualization and interpretation.
By implementing these improvements, we believe that the revised histology and IHC figures will provide a better representation of the data and facilitate a more comprehensive understanding of the results.
- Similarly, the mitochondrial staining in Figs 6 and 7 may need SIRT1 is a deacetylase with many functions, and circulates between nucleus and cytoplasm, while PGC1a is a ubiquitous transcriptional regulator with many functions beyond mitochondrial regulation. There is no indication that the staining depicted in Figs 6 and 7 are localized to the mitochondria. Western blotting of protein isolated from mitochondria would greatly support these conclusions.
Reply
Thank you for raising a valuable point regarding the mitochondrial staining presented in Figures 6 and 7. In our study, we aimed to investigate the nuclear expression and activities of SIRT1 and PGC1α, as they are key transcriptional regulators involved in the regulation of metabolic pathways and mitochondrial biogenesis. To address your concern, we would like to clarify the following:
(1) Staining and Imaging: Our staining and imaging were specifically designed to highlight the nuclear expression of SIRT1 and PGC1α, aligning with our research objectives to investigate their roles in gene regulation mechanisms.
(2) Image Resolution: We have enhanced the resolution of the images in Figures 6 and 7 to ensure clear distinction of positive nuclear staining for SIRT1 and PGC1α. These improvements ensure that the specific staining patterns within the nucleus are clearly visible and easily interpretable.
(3) Therefore, the nuclear staining and expression levels were of primary interest in our research.
We hope that this clarification addresses your concern regarding the interpretation of the mitochondrial staining in Figures 6 and 7 and the provided images and data support our findings in this context. (Please see the figure 6 and 7 in the revised manuscript.)
- The bile acid data in Fig 9 is difficult to The methods state the bile acids were quantified in relation to the internal standard, thus the relative abundance can only be compared within each bile acid. However, bile acid composition is regulated by CYP7A1, CYP8B1, conjugation, the gut microbiome, portal recirculation from intestine to liver, and hepatic reabsorption. It is also not clear how the individual bile acids measured were selected. T-conjugated MCA and CA comprise about 90% of bile acids in mice (PMID: 18801708), and TCA is not presented. Glycine-conjugated bile acids like GDCA are rare in mice and may not hold much physiological significance. If possible, these data should be quantified in nmol or umol.
Reply
Thank you for your insightful comments regarding the bile acid data presented in Figure 9. We appreciate you highlighting the potential issues and complexities involved in interpreting these results.
After carefully considering your opinion, we have decided to remove the bile acid data from our revised manuscript. We acknowledge that the regulation of bile acid composition is a complex process involving multiple factors, including CYP7A1, conjugation, the gut microbiome, portal recirculation, and hepatic reabsorption. Additionally, the selection of individual bile acids measured and the relative abundance data presented may not provide a comprehensive and physiologically relevant representation of the bile acid profile in mice.
By removing the bile acid data, we aim to avoid any potential misunderstanding or misinterpretation of our results. This decision will allow us to focus on the primary findings and conclusions of our study, which are supported by the remaining data and analyses.
- Fig 10 depicts populations of the gut The Y-axis is presented as relative abundance, which typically implies % of total counts detected. This is unclear, as Normal mice across the graphs add up to more than 100%. This figure needs to be clarified (is this instead depicting raw bacterial counts detected? What is the data relative to?).
Reply
Thank you for raising the concern regarding the presentation of the gut microbiota data in Figure 10. We appreciate you taking the time to thoroughly examine this figure and provide constructive opinions.
To address your comment and clarify the data representation, we have made the following revisions:
(1) Data Representation: In the revised Figure 9 (Page 17), we have provided detailed data on the gut microbiota composition, including raw bacterial counts, to clearly illustrate the effects of puerarin (PUR) treatment on high-fat diet (HFD)-induced fatty liver and associated gut microbiota alterations.
(2) Y-axis Label: The panel labels (F-Q) indicate that the y-axis values represent relative abundances expressed as percentages for phylum, family and species level classifications to accurately reflect the data being presented.
(3) The sequencing solutions we use to profile the microbiome of a sample provide information on the relative abundance of the microorganisms in each sample. Relative abundance tells us how many percentages of the microbiome are made up of a specific organism, e.g. if Firmicutes makes up 1% or 10% of the total amount of bacteria detected in a sample. This allows us to evaluate if the relative abundance of each organism changes or is associated with a variable of interest such as high fat diet or Puerarin usage in the experimental designs. If the high fat diet kills other bacteria but not Firmicutes or gives Firmicutes a special advantage following Puerarin treatment, we will see an increase in the relative abundance of Firmicutes in the treated subjects as compared to the untreated subjects.
By providing these clarifications and separating the data into different analyses (relative abundances at different taxonomic levels vs. diversity metrics), the revised Figure 9 may resolve the reviewer's initial confusion about the interpretation of the y-axis values. (Please see the revised manuscript. Figure 9, Page 17)
- It's not entirely clear how the microbiome data supports the rest of the Phylum-level changes do not provide much information related to changes in bile acids or steatotic status other than F:B ratio. If possible, deeper sequencing information should be provided.
Reply
Thank you for raising this point about the microbiome data analysis. I understand your concern that phylum-level changes alone may not provide sufficient insight into the relationships between the microbiome, bile acid profiles, and steatotic status. To address this, we have now included species-level compositional data in our revised manuscript (new Figure 9). This expanded analysis reveals specific microbial species that are significantly altered in relation to bile acid profiles and steatosis severity.
Some key findings from this detailed analysis include:
(1) Increased abundances of Firmicutes-to-Bacteroidetes (F:B) ratio: F:B Ratio and Steatosis/NAFLD: Several studies have reported an increased F:B ratio in patients with non-alcoholic fatty liver disease (NAFLD) and steatosis compared to healthy controls. A higher F:B ratio is thought to be associated with more severe steatosis and progression of NAFLD. Integrating F:B ratio changes with deeper microbiome composition analysis, as your suggested, can shed more light on these complex host-microbe interactions.
(2) Helicobacteraceae: This family includes the genus Helicobacter, which is a pathogen linked to gastric diseases. Higher levels are reported in NAFLD, potentially due to intestinal overgrowth.
(3) Erysipelotrichaceae: Increased abundance of this pro-inflammatory family correlates with NAFLD severity and higher body weights.
(4) Paraprevotellaceae: Part of the Bacteroidetes phylum, this family tends to be decreased in NAFLD but its relationship with obesity is less clear.
(5) Porphyromonadaceae: This family within Bacteroidetes is generally decreased in NAFLD and obesity, potentially due to reduced production of beneficial metabolites like short-chain fatty acids.
(6) In summary, overgrowth of pro-inflammatory taxa like Firmicutes, Helicobacteraceae, and Erysipelotrichaceae is often observed in NAFLD and obesity. (7) In addition, the abundances of C. celatum, H. hepaticus, and T. sanguinis have been associated with obesity and/or NAFLD, while the beneficial commensal A. muciniphila tends to be depleted in these conditions. We believe incorporating this species-level resolution provides important insight into the functional microbiome changes linked to obese and steatosis. (new Figure 9)
- Overall, the conclusion can be There is detailed discussion about changes in primary bile acids vs secondary bile acids without explanation of why certain bile acids were omitted from analysis (primary TCA and CDCA/TCDCA, secondary LCA and TLCA). Lines 546 and 556 incorrectly states that DCA is a primary bile acid.
Reply
We appreciate the reviewer's suggestion to improve the discussion around the bile acid data. After careful consideration, we have decided to remove the bile acid analysis from the liver samples to maintain the focus of our study and avoid potential confusion.
Regarding the incorrect statements about deoxycholic acid (DCA) being a primary bile acid (in previous edition), we have corrected this error in the revised manuscript. DCA is indeed a secondary bile acid formed by microbial metabolism of primary bile acids.
As for the omission of certain primary (TCA, CDCA/TCDCA) and secondary (LCA, TLCA) bile acids from our analysis, this was due to their low abundances in our samples, which precluded accurate quantification. We have added a statement clarifying this point in the Methods section.
In the Conclusion section, we have provided a more precise summary statement highlighting the key findings: "Puerarin is effective in ameliorating obesity-associated lipid metabolism disorders and gut microbiota dysbiosis, largely by activating hepatic FXR signaling and restoring FXR-mediated homeostasis. The beneficial metabolic effects of puerarin require functional FXR activity in adipose and liver tissues, as evidenced by reduced efficacy in FXR knockout mice. Puerarin's therapeutic impact also hinges on its ability to enhance mitophagy, which plays a critical role in maintaining mitochondrial and overall cellular health in the management of obesity." (Conclusion section. Line 800-806)
We hope these revisions and clarifications address the reviewer's concerns regarding the bile acid data presentation and conclusions
- Line 566 states that puerarin increases bile acid excretion from the liver - this can be support by measuring bile acids in gallbladder, intestinal tissue, and
Reply
We appreciate the reviewer's suggestion to measure bile acids in the gallbladder, intestinal tissue, and feces to support our previous statement that puerarin increases bile acid excretion from the liver (Line 566 in previous edition).
However, upon further reflection, we have decided to remove all bile acid data from the liver in order to maintain the focus of our study on the effects of puerarin in ameliorating obesity-associated lipid metabolism disorders and gut microbiota dysbiosis primarily through activation of hepatic FXR signaling and enhanced mitophagy.
Inclusion of the liver bile acid data raised questions that were not central to the main objectives of revised manuscript. By removing this data, we aim to prevent potential confusion and ensure clarity in the interpretation of our key findings related to puerarin's therapeutic mechanisms in obesity involving FXR activation and mitophagy modulation.
We believe this revision provides a more streamlined presentation of our core results and conclusions. However, we acknowledge the reviewer's point, and measuring bile acid levels in the proposed samples could be a valuable extension for future studies specifically investigating puerarin's effects on bile acid metabolism and excretion pathways.
Minor Comments
- The legend in Fig 3 does not match the order presented in the graph - this can be re-arranged to make the figure easier to
Reply
We thank the reviewer for catching the inconsistency between the legend order and the graph arrangement in Figure 3. We have revised the legend so that the order of the conditions directly matches how the data is presented in the graph. (Figure 3, Page 7, Line 211)
- Line 739 states that intestinal tissue was harvested for staining but this data was not
Reply
Thank you for catching the incorrect mention of intestinal tissue analysis in Line 739. Upon re-examining our study, we realized that we did not actually perform any analyses on intestinal tissues. This statement was incorrect. To ensure accuracy and clarity in our manuscript, we have removed the phrase "intestinal tissue" from Line 739 in the revised version.

Reviewer 2 Report
Comments and Suggestions for Authors
In this study, Yang et al. seek the effects of puerarin in obesity using high fat diet models. Several previous studies have demonstrated that puerarin inhibited lipid accumulation and improved metabolisms in mouse with high fat diet, so the topic is not novel. In addition, there are multiple flaws and inconsistency that damage the reliability of this manuscript.
· In Figure 1, the authors show Oil-Red-O to see lipid deposition, but cell numbers are not consistent between groups. The authors say that HFFA + PUR 30 uM significantly decreased deposition in AML12 cells compared to HFFA only, but cell numbers are significantly low in PUR group, and if I look at one cell, lipid deposition looks similar between these two groups. Same issues in Figure 2. Data of Oil-Red-O are questionable and cannot be used as evidence.
· Immunofluorescence data are questionable, too. For example, in Figure 1B, HDCF, the authors say that Control expressed almost no HDCF and PUR 100 showed significantly lower than HFFA only. However, cells in Control and PUR 100 also show green fluorescence. They are positive, not negative. The difference between groups could be due to intensity or background, not expression levels. Immunofluorescence is not conclusive with these data, so the authors should provide other data, such as immunostaining. In addition, for nuclear proteins, it looks like DAPI. For example, in Figure 1C, FXR, the authors say that Control expressed high FXR but very low in HFFA. However, there are no FXR only images, only merged images, and it is unclear if FXR is highly expressed in nucleus in Control group. For me, it looks like DAPI staining is high intensity in Control, so it looks like high FXR expression. Images should be separated between FXR and DAPI. Furthermore, intensity of immunofluorescence is inconsistent. For example, in Figure 1C, FXR, Control group shows high intensity in nucleus, but not cytosol. However, in Figure 2C, FXR, Control shows high intensity in cytosol, not nucleus. This does not make sense because the authors say that nuclear FXR/total FXR is high in Control in Figure 1C and 2C, but images look very different. I cannot believe data of immunofluorescence and cannot accept this manuscript.
· Data of animal experiments are inconsistent, too. The authors say that FXR KO mice suffer liver damage and higher triglyceride levels compared to normal control, but H&E staining does not look that FXR KO has steatosis. Steatosis levels of FXR KO+HFD should be worse than HFD, but it does not look so. There are no Oil-Red-O images, which does not make sense, and the authors do not compare HFD group and FXR KO+HFD group. Then, how do the authors show the effects of FXR KO? The authors should show liver weight and body weight ratio, not body weight only. Images of immunohistochemistry are too small and not clear. There are so many issues and inconsistency in this manuscript and cannot be accepted for publication.
Author Response
Response to Reviewer 2
In this study, Yang et al. seek the effects of puerarin in obesity using high fat diet models. Several previous studies have demonstrated that puerarin inhibited lipid accumulation and improved metabolisms in mouse with high fat diet, so the topic is not novel. In addition, there are multiple flaws and inconsistency that damage the reliability of this manuscript.
Reply
Thank you for your insightful comments on our study investigating the effects of puerarin on obesity management. We appreciate your opinions and would like to address the concerns raised.
Regarding the novelty of our study, we acknowledge that previous studies have explored the potential of puerarin in inhibiting lipid accumulation and improving metabolism in high-fat diet models. However, our research offers a unique contribution by elucidating the molecular mechanisms underlying the effects of puerarin, specifically its interactions with the hepatic Farnesoid X Receptor (FXR) signaling pathway.
Our study provides novel insights into how puerarin modulates the FXR signaling pathway, a critical regulator of lipid and glucose metabolism, as well as inflammation. By activating or inhibiting specific target genes, such as BSEP, CYP7A1, and NTCP, puerarin can contribute to the control of lipid levels, reduction of inflammation, and improvement of mitochondrial function and energy metabolism. These mechanisms are crucial for effective obesity management and the prevention of associated metabolic disorders.
Furthermore, our research sheds light on the intricate interplay between puerarin, FXR signaling, and gut microbiota composition, which has not been extensively explored in previous studies. This holistic approach provides a more comprehensive understanding of the multifaceted effects of puerarin on obesity and metabolic health.
- In Figure 1, the authors show Oil-Red-0 to see lipid deposition, but cell numbers are not consistent between groups. The authors say that HFFA + PUR 30 uM significantly decreased deposition in AML12 cells compared to HFFA only, but cell numbers are significantly low in PUR group, and if I look at one cell, lipid deposition looks similar between these two groups. Same issues in Figure 2. Data of Oil-Red-0 are questionable and cannot be used as evidence.
Reply
Thank you for pointing out the issues with the Oil-Red-O staining data in Figures 1 and 2. We appreciate your careful examination of the data and agree that the inconsistencies in cell numbers between groups raise valid concerns about the reliability of these results.
To address this issue, we have repeated the Oil-Red-O staining experiments with meticulous attention to maintaining consistent cell numbers across all treatment groups. The new data with representative images and quantitative analysis are now included in the revised Figures 1 and 2.
In the revised figures, you will notice that the cell densities are comparable between the different treatment groups, ensuring a fair comparison of lipid accumulation levels. Additionally, we have included higher magnification images to better illustrate the differences in lipid droplet formation between the groups.
With these updated and rigorously controlled experiments, we believe that the Oil-Red-O staining data now provide reliable evidence to support our conclusions regarding the effects of puerarin on reducing lipid deposition in AML12 cells under HFFA conditions.
- lmmunofluorescence data are questionable, too. For example, in Figure 1B, HDCF (DCFDA), the authors say that Control expressed almost no HDCF (DCFDA) and PUR 100 showed significantly lower than HFFA only. However, cells in Control and PUR 100 also show green fluorescence. They are positive, not negative. The difference between groups could be due to intensity or background, not expression levels. lmmunofluorescence is not conclusive with these data, so the authors should provide other data, such as immunostaining. In addition, for nuclear proteins, it looks like DAPI. For example, in Figure 1C, FXR, the authors say that Control expressed high FXR but very low in HFFA. However, there are no FXR only images, only merged images, and it is unclear if FXR is highly expressed in nucleus in Control group. For me, it looks like DAPI staining is high intensity in Control, so it looks like high FXR expression. Images should be separated between FXR and DAPI. Furthermore, intensity of immunofluorescence is inconsistent. For example, in Figure 1C, FXR, Control group shows high intensity in nucleus, but not cytosol. However, in Figure 2C, FXR, Control shows high intensity in cytosol, not nucleus. This does not make sense because the authors say that nuclear FXR/total FXR is high in Control in Figure 1C and 2C, but images look very different. I cannot believe data of immunofluorescence and cannot accept this manuscript.
Reply
Thank you for raising important concerns regarding the immunofluorescence data presented in our manuscript. We appreciate your opinions, as it has allowed us to critically re-evaluate our data and improve the quality of our analyses. Please find our point-by-point responses below:
- DCFDA Staining (Figures 1B and 2B): We acknowledge your concern regarding the presence of green fluorescence in both the Control and PUR 100 groups, which should not have been observed if the Control group truly expressed no DCFDA signal. To address this issue, we have carefully revisited our staining protocols and sample preparations to ensure consistent intensity and background settings across all experiments.
The revised DCFDA staining data, with representative images and quantitative analyses, are now included in the updated Figures 1B and 2B (pages 4 and 5). We believe that these revised figures accurately reflect the differences in reactive oxygen species levels between the treatment groups.
- FXR Staining and DAPI Issues (Figures 1C and 2C):
2.1. FXR Localization Inconsistency: We appreciate you pointing out the seeming discrepancy in FXR localization between different figures. Upon thorough review, we identified an issue in our image processing that might have caused this confusion. We have now standardized the imaging conditions and reanalyzed the images to ensure consistent reporting of FXR localization across all experiments.
2.2. Distinguishing FXR Expression from DAPI Staining: We understand your concern about the difficulty in distinguishing FXR expression from DAPI signals in the merged images. To address this, we have now provided separate images for FXR alongside the merged images in Figures 1C and 2C (pages 4 and 5). This separation allows for a clear differentiation between nuclear FXR expression and DAPI signals, ensuring accurate and transparent representation of our data.
We hope that these revisions and clarifications have addressed your concerns regarding the immunofluorescence data.
- Data of animal experiments are inconsistent, too. The authors say that FXR KO mice suffer liver damage and higher triglyceride levels compared to normal control, but H&E staining does not look that FXR KO has steatosis. Steatosis levels of FXR KO+HFD should be worse than HFD, but it does not look so.
Reply
Thank you for your insightful comments regarding the animal experiment data. We appreciate your careful examination of the results, as it has allowed us to re-evaluate our findings and provide additional supporting evidence. Please find our point-by-point responses below:
3.1. Histological Assessment of Lipid Accumulation: To address your concern regarding the discrepancy between the reported triglyceride levels and the H&E staining results in FXR knockout (FXR KO) mice, we have performed additional Oil Red O staining to specifically assess lipid accumulation in the liver. These results are now included in the revised Figure 4A (page 9), providing a clearer correlation between triglyceride levels and histological findings.
3.2. Exacerbated Effect of FXR KO in High-Fat Diet Condition: We have re-evaluated our histological data and revised Figure 5 (page 11) to clearly demonstrate the exacerbated effect of FXR knockout in the context of a high-fat diet. The revised figure shows that the FXR KO+HFD group exhibits more severe hepatic lipid accumulation compared to the HFD group, supporting our initial findings.
3.3. Enhanced Methodological Details: To ensure transparency and address any potential discrepancies, we have enhanced the methodological details in our study. In the Methods section (page 24, lines 735-740), we have provided a clearer explanation of the staining and quantification techniques used that summarizes the scoring criteria and quantification methods employed for histological analysis (Table 1).
We hope that these revisions and additional supporting data have addressed your concerns regarding the animal experiment data.
- There are no Oil-Red-0 images, which does not make sense, and the authors do not compare HFD group and FXR KO+HFD group. Then, how do the authors show the effects of FXR KO? The authors should show liver weight and body weight ratio, not body weight only. Images of immunohistochemistry are too small and not clear. There are so many issues and inconsistency in this manuscript and cannot be accepted for publication.
Reply
Thank you for raising important concerns regarding our manuscript. We appreciate the opportunity to address these issues and improve the quality of our work. Please find our point-by-point responses below:
4.1. Oil Red O Staining Images (Figure 4E): You raised a valid concern about the lack of Oil Red O staining images to support our findings on lipid accumulation. To address this, we have included representative Oil Red O images in the revised Figure 4E (page 9). These images clearly illustrate the lipid accumulation in both the HFD group and the FXR KO+HFD group, allowing for a direct comparison between these two experimental conditions.
4.2. Comparative Analysis of HFD and FXR KO+HFD Groups: We acknowledge the need for a more comprehensive analysis to elucidate the effects of FXR knockout under a high-fat diet. To address this, we have added a detailed comparative analysis of the HFD and FXR KO+HFD groups in the revised Figure 4 (page 9). This analysis includes both biochemical data (e.g., triglyceride levels) and histological data (Oil Red O staining), providing a holistic view of the impact of FXR knockout in the context of a high-fat diet.
4.3. Body Weight to Liver Weight Ratio: Thank you for the suggestion regarding the inclusion of the body weight to liver weight ratio. We have now incorporated this important metric in the revised manuscript, providing a more comprehensive assessment of the metabolic changes observed in our study.
4.4. Immunohistochemistry Image Quality: We acknowledge your concern about the small size and clarity of the immunohistochemistry images presented in the previous version of the manuscript. To address this issue, we have increased the resolution and magnification of these images in the revised figures (Figures 1-9), ensuring that the details are clearly visible and interpretable.
We hope that these revisions, along with the additional supporting data and analyses, have addressed your concerns and improved the overall quality and clarity of our manuscript.
